# WHY RESAMPLING OUTPERFORMS REWEIGHTING FOR CORRECTING SAMPLING BIAS WITH STOCHASTIC GRADIENTS

**Jing An, Lexing Ying and Yuhua Zhu**
Stanford University
{jingan, lexing, yuhuazhu}@stanford.edu

## ABSTRACT

A data set sampled from a certain population is biased if the subgroups of the population are sampled at proportions that are significantly different from their underlying proportions. Training machine learning models on biased data sets requires correction techniques to compensate for the bias. We consider two commonly-used techniques, resampling and reweighting, that rebalance the proportions of the subgroups to maintain the desired objective function. Though statistically equivalent, it has been observed that resampling outperforms reweighting when combined with stochastic gradient algorithms. By analyzing illustrative examples, we explain the reason behind this phenomenon using tools from dynamical stability and stochastic asymptotics. We also present experiments from regression, classification, and off-policy prediction to demonstrate that this is a general phenomenon. We argue that it is imperative to consider the objective function design and the optimization algorithm together while addressing the sampling bias.

## 1 INTRODUCTION

A data set sampled from a certain population is called *biased* if the subgroups of the population are sampled at proportions that are significantly different from their underlying population proportions. Applying machine learning algorithms naively to biased training data can raise serious concerns and lead to controversial results (Sweeney, 2013; Kay et al., 2015; Menon et al., 2020). In many domains such as demographic surveys, fraud detection, identification of rare diseases, and natural disasters prediction, a model trained from biased data tends to favor oversampled subgroups by achieving high accuracy there while sacrificing the performance on undersampled subgroups. Although one can improve by diversifying and balancing during the data collection process, it is often hard or impossible to eliminate the sampling bias due to historical and operational issues.

In order to mitigate the biases and discriminations against the undersampled subgroups, a common technique is to preprocess the data set by compensating the mismatch between *population proportion* and the *sampling proportion*. Among various approaches, two commonly-used choices are *reweighting* and *resampling*. In reweighting, one multiplies each sample with a ratio equal to its population proportion over its sampling proportion. In resampling, on the other hand, one corrects the proportion mismatch by either generating new samples for the undersampled subgroups or selecting a subset of samples for the oversampled subgroups. Both methods result in statistically equivalent models in terms of the loss function (see details in Section 2). However, it has been observed in practice that resampling often outperforms reweighting significantly, such as boosting algorithms in classification (Galar et al., 2011; Seiffert et al., 2008), off-policy prediction in reinforcement learning (Schlegel et al., 2019) and so on. The obvious question is why.

**Main contributions.** Our main contribution is to provide an answer to this question: resampling outperforms reweighting because of the stochastic gradient-type algorithms used for training. To the best of our knowledge, our explanation is the first theoretical quantitative analysis for this phenomenon. With stochastic gradient descent (SGD) being the dominant method for model training, our analysis is based on some recent developments for understanding SGD. We show via simple and

explicitly analyzable examples why resampling generates expected results while reweighting performs undesirably. Our theoretical analysis is based on two points of view, one from the dynamical stability perspective and the other from stochastic asymptotics.

In addition to the theoretical analysis, we present experimental examples from three distinct categories (classification, regression, and off-policy prediction) to demonstrate that resampling outperforms reweighting in practice. This empirical study illustrates that this is a quite general phenomenon when models are trained using stochastic gradient type algorithms.

Our theoretical analysis and experiments show clearly that adjusting only the loss functions is not sufficient for fixing the biased data problem. The output can be disastrous if one overlooks the optimization algorithm used in the training. In fact, recent understanding has shown that objective function design and optimization algorithm are closely related, for example optimization algorithms such as SGD play a key role in the generalizability of deep neural networks. Therefore in order to address the biased data issue, we advocate for *considering data, model, and optimization as an integrated system.*

**Related work.** In a broader scope, resampling and reweighting can be considered as instances of preprocessing the training data to tackle biases of machine learning algorithms. Though there are many well-developed resampling (Mani & Zhang, 2003; He & Garcia, 2009; Maciejewski & Stefanowski, 2011) and reweighting (Kumar et al., 2010; Malisiewicz et al., 2011; Chang et al., 2017) techniques, we only focus on the reweighting approaches that do not change the optimization problem. It has been well-known that training algorithms using disparate data can lead to algorithmic discrimination (Bolukbasi et al., 2016; Caliskan et al., 2017), and over the years there have been growing efforts to mitigate such biases, for example see (Amini et al., 2019; Kamiran & Calders, 2012; Calmon et al., 2017; Zhao et al., 2019; López et al., 2013). We also refer to (Guo et al., 2017; He & Ma, 2013; Krawczyk, 2016) for a comprehensive review of this growing research field.

Our approaches for understanding the dynamics of resampling and reweighting under SGD are based on tools from numerical analysis for stochastic systems. Connections between numerical analysis and stochastic algorithms have been rapidly developing in recent years. The dynamical stability perspective has been used in (Wu et al., 2018) to show the impact of learning rate and batch size in minima selection. The stochastic differential equations (SDE) approach for approximating stochastic optimization methods can be traced in the line of work (Li et al., 2017; 2019; Rotskoff & Vanden-Eijnden, 2018; Shi et al., 2019), just to mention a few.

## 2 PROBLEM SETUP

Let us consider a population that is comprised of two different groups, where a proportion $a_1$ of the population belongs to the first group, and the rest with the proportion $a_2 = 1 - a_1$ belongs to the second (i.e., $a_1, a_2 > 0$ and $a_1 + a_2 = 1$). In what follows, we shall call $a_1$ and $a_2$ the *population proportions*. Consider an optimization problem for this population over a parameter $\theta$. For simplicity, we assume that each individual from the first group experiences a loss function $V_1(\theta)$, while each individual from the second group has a loss function of type $V_2(\theta)$. Here the loss function $V_1(\theta)$ is assumed to be identical across all members of the first group and the same for $V_2(\theta)$ across the second group, however it is possible to extend the formulation to allow for loss function variation within each group. Based on this setup, a minimization problem over the whole population is to find

$$\theta^* = \arg\min_\theta V(\theta), \quad \text{where } V(\theta) \equiv a_1 V_1(\theta) + a_2 V_2(\theta). \tag{1}$$

For a given set $\Omega$ of $N$ individuals sampled uniformly from the population, the empirical minimization problem is

$$\theta^* = \arg\min_\theta \frac{1}{N} \sum_{r \in \Omega} V_{i_r}(\theta), \tag{2}$$

where $i_r \in \{1, 2\}$ denotes which group an individual $r$ belongs to. When $N$ grows, the empirical loss in (2) is consistent with the population loss in (1) as there are approximately $a_1$ fraction of samples from the first group and $a_2$ fraction of samples from the second.

However, the sampling can be far from uniformly random in reality. Let $n_1$ and $n_2$ with $n_1+n_2 = N$ denote the number of samples from the first and the second group, respectively. It is convenient to define $f_i, i = 1, 2$ as the sampling proportions for each group, i.e., $f_1 = n_1/N$ and $f_2 = n_2/N$ with $f_1 + f_2 = 1$. The data set is biased when the sampling proportions $f_1$ and $f_2$ are different from the population proportions $a_1$ and $a_2$. In such a case, the empirical loss is $f_1V_1(\theta) + f_2V_2(\theta)$, which is clearly wrong when compared with (1).

Let us consider two basic strategies to adjust the model: reweighting and resampling. In reweighting, one assigns to each sample $r \in \Omega$ a weight $a_{i_r}/f_{i_r}$ and the reweighting loss function is

$$V_w(\theta) \equiv \frac{1}{N} \sum_{r \in \Omega} \frac{a_{i_r}}{f_{i_r}} V_{i_r}(\theta) = a_1 V_1(\theta) + a_2 V_2(\theta). \tag{3}$$

In resampling, one either adds samples to the minority group (i.e., oversampling) or removing samples from the majority group (i.e., undersampling). Although the actual implementation of oversampling and undersampling could be quite sophisticated in order to avoid overfitting or loss of information, mathematically we interpret the resampling as constructing a new set of samples of size $M$, among which $a_1 M$ samples are of the first group and $a_2 M$ samples of the second. The resampling loss function is

$$V_s(\theta) \equiv \frac{1}{M} \sum_s V_{i_s}(\theta) = a_1 V_1(\theta) + a_2 V_2(\theta). \tag{4}$$

Notice that both $V_w(\theta)$ and $V_s(\theta)$ are consistent with the population loss function $V(\theta)$. This means that, under mild conditions on $V_1(\theta)$ and $V_2(\theta)$, a deterministic gradient descent algorithm from a generic initial condition converges to similar solutions for $V_w(\theta)$ and $V_s(\theta)$. For a stochastic gradient descent algorithm, the expectations of the stochastic gradients of $V_w(\theta)$ and $V_s(\theta)$ also agree at any $\theta$ value. However, as we shall explain below, the training behavior can be drastically different for a stochastic gradient algorithm. The key reason is that the variances experienced for $V_w(\theta)$ and $V_s(\theta)$ can be drastically different: computing the variances of gradients for resampling and reweighting reveals that

$$\mathbb{V}\left[\nabla \hat{V}_s(\theta)\right] = a_1 \nabla V_1(\theta)\nabla V_1(\theta)^T + a_2 \nabla V_2(\theta)\nabla V_2(\theta)^T - (\mathbb{E}[\nabla \hat{V}_s(\theta)])^2,$$

$$\mathbb{V}\left[\nabla \hat{V}_w(\theta)\right] = \frac{a_1^2}{f_1} \nabla V_1(\theta)\nabla V_1(\theta)^T + \frac{a_2^2}{f_2} \nabla V_2(\theta)\nabla V_2(\theta)^T - (\mathbb{E}[\nabla \hat{V}_w(\theta)])^2. \tag{5}$$

These formulas indicate that, when $f_1/f_2$ is significantly misaligned with $a_1/a_2$, the variance of reweighting can be much larger. Without knowing the optimal learning rates a priori, it is difficult to select an efficient learning rate for reliable and stable performance for stiff problems, when only reweighting is used. In comparison, resampling is more favorable especially when the choice of learning rates is restrictive.

## 3 STABILITY ANALYSIS

Let us use a simple example to illustrate why resampling outperforms reweighting under SGD, from the viewpoint of stability. Consider two loss functions $V_1$ and $V_2$ with disjoint supports,

$$V_1(\theta) = \begin{cases} \frac{1}{2}(\theta + 1)^2 - \frac{1}{2}, & \theta \leq 0 \\ 0, & \theta > 0, \end{cases} \quad V_2(\theta) = \begin{cases} 0, & \theta \leq 0 \\ \frac{1}{2}(\theta - 1)^2 - \frac{1}{2}, & \theta > 0, \end{cases} \tag{6}$$

each of which is quadratic on its support. The population loss function is $V(\theta) = a_1 V_1(\theta) + a_2 V_2(\theta)$, with two local minima at $\theta = -1$ and $\theta = 1$. The gradients for $V_1$ and $V_2$ are

$$\nabla V_1(\theta) = \begin{cases} \theta + 1, & \theta \leq 0 \\ 0, & \theta > 0. \end{cases}, \quad \nabla V_2(\theta) = \begin{cases} 0, & \theta \leq 0 \\ \theta - 1, & \theta > 0. \end{cases}$$

Suppose that the population proportions satisfy $a_2 > a_1$, then $\theta = 1$ is the global minimizer and it is desired that SGD should be stable near it. However, as shown in Figure 1, when the sampling proportion $f_2$ is significantly less than the population proportion $a_2$, for reweighting $\theta = 1$ can easily become unstable: even if one starts near the global minimizer $\theta = 1$, the trajectories for reweighting

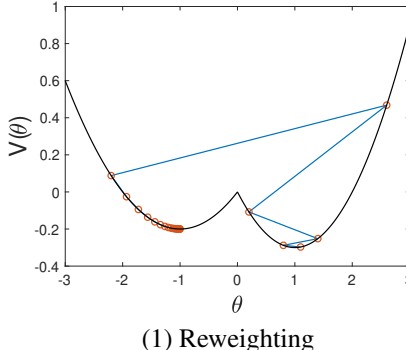
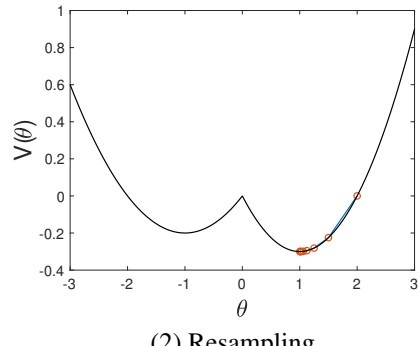

(1) Reweighting          (2) Resampling

Figure 1: Comparison of reweighting and resampling with $a_1/a_2 = 0.4/0.6$ and $f_1/f_2 = 0.9/0.1$ at the learning rate $\eta = 0.5$. The resampling strategy here is to randomly select the sub-population $i$ with the probability $a_i$ with replacement in each iteration. (1) For reweighting, the trajectory starting from $\theta_0 = 1.1$ can end up at $\theta = -1$ after a few iterations, but $\theta = -1$ is not the global minimizer. (2) For resampling, the trajectory starting from $\theta_0 = 2.0$ stays close to the desired minimizer $\theta = 1$. Hence resampling is more reliable than reweighting. We include more comparisons with various learning rates in Appendix D to show that resampling is stable for a wider range of $\eta$.

always gear towards $\theta = -1$ after a few steps (see Figure 1(1)). On the other hand, for resampling $\theta = 1$ is quite stable (see Figure 1(2)).

The expectations of the stochastic gradient are the same for both methods. It is the difference in the second moment that explains why trajectories near the two minima exhibit different behaviors. Our explanation is based on the stability analysis framework used in (Wu et al., 2018). By definition, a stationary point $\theta^*$ is *stochastically stable* if there exists a uniform constant $0 < C \leq 1$ such that $\mathbb{E}[\|\theta_k - \theta^*\|^2] \leq C\|\theta_0 - \theta^*\|^2$, where $\theta_k$ is the $k$-th iterate of SGD. The stability conditions for resampling and reweighting are stated in the following two lemmas, in which we use $\eta$ to denote the learning rate.

**Lemma 1.** *For resampling, the conditions for the SGD to be stochastically stable around $\theta = -1$ and $\theta = 1$ are respectively*

$$(1 - \eta a_1)^2 + \eta^2 a_1 a_2 \leq 1, \quad (1 - \eta a_2)^2 + \eta^2 a_1 a_2 \leq 1.$$

**Lemma 2.** *For reweighting, the condition for the SGD to be stochastically stable around $\theta = -1$ and $\theta = 1$ are respectively*

$$(1 - \eta a_1)^2 + \eta^2 f_1 f_2 \left(\frac{a_1}{f_1}\right)^2 \leq 1, \quad (1 - \eta a_2)^2 + \eta^2 f_1 f_2 \left(\frac{a_2}{f_2}\right)^2 \leq 1.$$

Note that the stability conditions for resampling are independent of the sampling proportions $(f_1, f_2)$, while the ones for reweighting clearly depend on $(f_1, f_2)$. We defer the detailed computations to Appendix A.

Lemma 2 shows that reweighting can incur a more stringent stability criterion. Let us consider the case $a_1 = \frac{1}{2} - \epsilon, a_2 = \frac{1}{2} + \epsilon$ with a small constant $\epsilon > 0$ and $f_2/f_1 \ll 1$. For reweighting, the global minimum $\theta = 1$ is stochastically stable only if $\eta(1 + f_1/f_2) \leq 4 + O(\epsilon)$. This condition becomes rather stringent in terms of the learning rate $\eta$ since $f_1/f_2 \gg 1$. On the other hand, the local minimizer $\theta = -1$ is stable if $\eta(1 + f_2/f_1) \leq 4 + O(\epsilon)$, which could be satisfied for a broader range of $\eta$ because $f_2/f_1 \ll 1$. In other words, for a fixed learning rate $\eta$, when the ratio $f_2/f_1$ between the sampling proportions is sufficiently small, the desired minimizer $\theta = 1$ is no longer statistically stable with respect to SGD.

## 4 SDE ANALYSIS

The stability analysis can only be carried for a learning rate $\eta$ of a finite size. However, even for a small learning rate $\eta$, one can show that the reweighting method is still unreliable from a different perspective. This section applies stochastic differential equation analysis to demonstrate it.

Let us again use a simple example to illustrate the main idea. Consider the following two loss functions,

$$V_1(\theta) = \begin{cases} |\theta + 1| - 1, & \theta \leq 0 \\ \epsilon\theta, & \theta > 0 \end{cases}, \quad V_2(\theta) = \begin{cases} -\epsilon\theta, & \theta \leq 0 \\ |\theta - 1| - 1, & \theta > 0 \end{cases},$$

with $0 < \epsilon \ll 1$. The population loss function is $V(\theta) = a_1 V_1(\theta) + a_2 V_2(\theta)$ with local minimizers $\theta = -1$ and $\theta = 1$. Note that the $O(\epsilon)$ terms are necessary. Without it, if the SGD starts in $(-\infty, 0)$, all iterates will stay in this region because there is no drift from $V_2(\theta)$. Similarly, if the SGD starts in $(0, \infty)$, no iterates will move to $(-\infty, 0)$. That means the result of SGD only depends on the initialization when $O(\epsilon)$ term is absent.

In Figure 2, we present numerical simulations of the resampling and reweighting methods for the designed loss function $V(\theta)$. If $a_2 > a_1$, then the global minimizer of $V(\theta)$ is $\theta = 1$ (see the Figure 2(1)). Consider a setup with population proportions $a_1/a_2 = 0.4/0.6$ along sampling proportions $f_1/f_2 = 0.9/0.1$, which are quite different. Figures 2(2) and (3) show the dynamics under the reweighting and resampling methods, respectively. The plots show that, while the trajectory for resampling is stable across time, the trajectory for reweighting quickly escapes to the (non-global) local minimizer $\theta = -1$ even when it starts near the global minimizer $\theta = 1$.

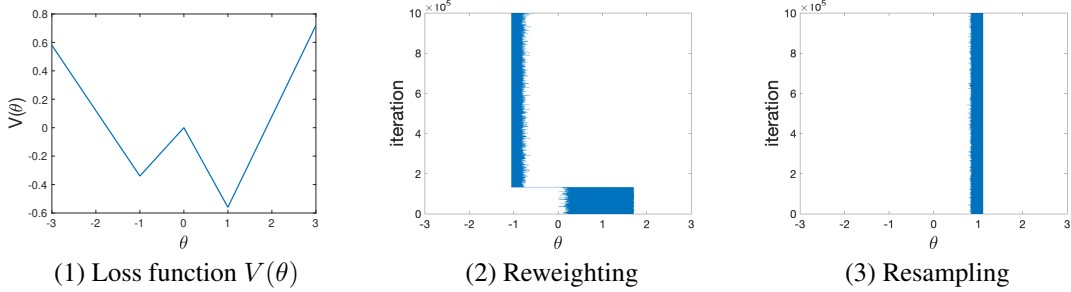

(1) Loss function $V(\theta)$      (2) Reweighting      (3) Resampling

Figure 2: Comparison of reweighting and resampling with learning rate $\eta = 0.12$. We set $a_1/a_2 = 0.4/0.6$, $f_1/f_2 = 0.9/0.1$ and $\epsilon = 0.1$. Both experiments start at $\theta_0 = 0.9$. The resampling strategy here is to randomly select the sub-population $i$ with the probability $a_i$ with replacement in each iteration. In (2) where reweighting is used, the trajectory skips to the local minimizer $\theta = -1$ later. In (3) where resampling is used, it stabilizes at the global minimizer $\theta = 1$ all the time. We include more comparisons with various learning rates in Appendix D to show that resampling is more reliable for a wider range of $\eta$.

When the learning rate is sufficiently small, one can approximate the SGD by an SDE, which in this piece-wise linear loss example is approximately a Langevin dynamics with a piecewise constant mobility. In particular when the dynamics reaches equilibrium, the stationary distribution of the stochastic process is approximated by a Gibbs distribution, which gives the probability densities at the stationary points. Let us denote $p_s(\theta)$ and $p_w(\theta)$ as the stationary distribution over $\theta$ under resampling and reweighting, respectively. Following lemmas quantitatively summarize the results.

**Lemma 3.** *When $a_2 > a_1$, $V(1) < V(-1)$. The stationary distribution for resampling satisfies the relationship*

$$\frac{p_s(1)}{p_s(-1)} = \exp\left(-\frac{2}{a_1 a_2 \eta}(V(1) - V(-1))\right) + O(\epsilon) > 1.$$

**Lemma 4.** *With $a_2 > a_1$, $V(1) < V(-1) < 0$. Under the condition $\frac{f_2}{f_1} \leq \frac{a_2}{a_1}\sqrt{\frac{V(-1)}{V(1)}}$ for the sampling proportions, the stationary distribution for reweighting satisfies the relationship*

$$\frac{p_w(1)}{p_w(-1)} = \frac{a_1^2/f_1^2}{a_2^2/f_2^2} \exp\left(-\frac{2f_2/f_1}{a_2^2\eta}V(1) + \frac{2f_1/f_2}{a_1^2\eta}V(-1)\right) + O(\epsilon) < 1.$$

The proofs of the above two lemmas can be found in Appendix B. Lemma 3 shows that for resampling it is always more likely to find $\theta$ at the global minimizer 1 than at the local minimizer $-1$. Lemma 4 states that for reweighting it is more likely to find $\theta$ at the local minimizer $-1$ when $\frac{f_2}{f_1} \leq \frac{a_2}{a_1}\sqrt{\frac{V(-1)}{V(1)}}$. Together, they explain the phenomenon shown in Figure 2.

To better understand the condition in Lemma 4, let us consider the case $a_1 = \frac{1}{2} - \epsilon, a_2 = \frac{1}{2} + \epsilon$ with a small constant $\epsilon > 0$. Under this setup, $V(-1)/V(1) \approx 1$. Whenever the ratio of the sampling proportions $f_2/f_1$ is significantly less than the ratio of the population proportions $a_2/a_1 \approx 1$, reweighting will lead to the undesired behavior. The smaller the ratio $f_2/f_1$ is, the less likely the global minimizer will be visited.

The reason for constructing the above piecewise linear loss function is to obtain an approximately explicitly solvable SDE with a constant coefficient for the noise. One can further extend the results in 1D for piecewise strictly convex function with two local minima (See Lemmas 9 and 10 in Appendix B.3). Here we present the most general results in 1D, that is, piecewise strictly convex function with finite number of local minima. One may consider the population loss function $V(\theta) = \sum_{i=1}^{k} a_i V_i(\theta)$ with $V_i(\theta) = h_i(\theta)$ for $\theta_{i-1} < \theta \le \theta_i$ and $V_i(\theta) = O(\epsilon)$ otherwise, where $h_i(\theta)$ are strictly convex functions and continuously differentiable, $O(\epsilon)$ term is sufficiently small and smooth. Here $\{\theta_i\}_{i=1}^{k-1}$ are $k-1$ disjoint points, and $\theta_0 = -\infty, \theta_k = \infty$. We assume that $V(\theta)$ has $k$ local minimizers $\theta_i^*$ for $\theta_i^* \in (\theta_{i-1}, \theta_i)$. We present the following two lemmas with suitable assumptions (See Appendix B.3 for details of assumptions, the proof and follow-up discussions).

**Lemma 5.** *The stationary distribution for resampling at any two local minizers $\theta_p^*, \theta_q^*$ with $p > q$ satisfies the relationship*

$$\frac{p_s(\theta_p^*)}{p_s(\theta_q^*)} = \exp\left[\frac{2}{\eta}\int_{\theta_q^*}^{\theta_p} \frac{1}{h_p'(\theta)}d\theta\left(\frac{1}{1-a_p} - \frac{1}{1-a_q}\right)\right] + O(\epsilon) = \begin{cases} > 1, & \text{if } a_p > a_q; \\ < 1, & \text{if } a_p < a_q, \end{cases}$$

**Lemma 6.** *The stationary distribution for reweighting at any two local minizers $\theta_p^*, \theta_q^*$ with $p > q$ satisfies the relationship*

$$\frac{p_w(\theta_p^*)}{p_w(\theta_q^*)} = \exp\left[\frac{2}{\eta}\int_{\theta_p^*}^{\theta_p} \frac{1}{h_p'(\theta)}d\theta\left(\frac{f_p}{a_p(1-f_p)} - \frac{f_q}{a_q(1-f_q)}\right)\right] + O(\epsilon).$$

**Multi-dimensional results.** Let us now consider the minimization of $V(\theta) = a_1 V_1(\theta) + a_2 V_2(\theta)$ for more general $V_1, V_2$ and also $\theta$ in high dimensions. It is in fact not clear how to extend the above stochastic analysis to more general functions $V(\theta)$. Instead we focus on the *transition time* from one stationary point to another in order to understand the behavior of resampling and reweighting. For this purpose, we again resort to the SDE approximation of the SGD in the continuous time limit.

Such a SDE approximation, first introduced in (Li et al., 2017), involves a data-dependent covariance coefficient for the diffusion term and is justified in the weak sense with an error of order $O(\sqrt{\eta})$. More specifically, the dynamics can be approximated by

$$d\Theta = -\nabla V(\Theta)dt + \sqrt{\eta}\Sigma(\Theta)^{1/2}dB, \tag{7}$$

where $\Theta(t = k\eta) \approx \theta_k$ for the step $k$ parameter $\theta_k$, $\eta$ is the learning rate, and $\Sigma(\Theta)$ is the covariance of the stochastic gradient at location $\Theta$. In the SDE theory, the drift term $\nabla V(\cdot)$ is usually assumed to be Lipschitz. However, in machine learning (for example neural network training with non-smooth activation functions), it is common to encounter non-Lipschitz gradients of loss functions (as in the example presented in Section 3). To fill this gap, we provide in Appendix C a justification of SDE approximation for the drift with jump discontinuities, based on the proof presented in (Müller-Gronbach et al., 2020). The following two lemmas summarize the transition times between the two local minimizers.

**Lemma 7.** *Assume that there are only two local minimizers $\theta_1^*, \theta_2^*$ for the objective function $V(\theta)$. Let $\tau_{\theta_1^* \to \theta_2^*}$ be the transition time for $\Theta(t)$ in (7) from the $\epsilon$-neighborhood of $\theta_1^*$ (a closed ball of radius $\epsilon$ centered at $\theta_1^*$) to the $\epsilon$-neighborhood of $\theta_2^*$ and $\tau_{\theta_2^* \to \theta_1^*}$ be the transition time in the opposite direction. Then*

$$\frac{\mathbb{E}[\tau_{\theta_1^* \to \theta_2^*}]}{\mathbb{E}[\tau_{\theta_2^* \to \theta_1^*}]} = \sqrt{\frac{\det(\nabla^2 L(\theta_2^*))}{\det(\nabla^2 L(\theta_1^*))}} \exp\left(\frac{2}{\eta}\left(\frac{\delta V(\theta_1^*)}{\Sigma(\theta_1^*)} - \frac{\delta V(\theta_2^*)}{\Sigma(\theta_2^*)}\right)\right) + O(\sqrt{\epsilon}).$$

*Here $\det(\nabla^2 L(\theta_1^*))$ and $\det(\nabla^2 L(\theta_2^*))$ are the determinants of the Hessians at $\theta_1^*$ and $\theta_2^*$, respectively. $\delta V(\theta_k^*) \equiv V(\theta^\circ) - V(\theta_k^*)$ for $k = 1, 2$ where $\theta^\circ$ is the saddle point between $\theta_1^*, \theta_2^*$.[1]*

---

[1] The formal definition of $\theta^\circ$: Let $\theta(t)$ be a path with $\theta(0) = \theta_1^*, \theta(1) = \theta_2^*$, then $\hat{\theta}(t) = \arg\min_{\theta(t)} \sup_{t \in [0,1]} V(\theta(t))$ is the path with minimal saddle point height among all continuous paths. $\theta^\circ = \sup_{t \in (0,1)} \hat{\theta}(t)$ is the saddle point of this path.

This lemma is known in the diffusion process literature as the Eyring-Kramers formula; see, e.g., (Berglund, 2011; Bovier et al., 2004; 2005). Using the above lemma, we obtain the following result for the transition times for resampling and reweighting.

**Lemma 8.** *Assume that there are only two local minimizers $\theta_1^*, \theta_2^*$ for the objective function $V(\theta)$. Also assume that the loss function $V_1(\cdot)$ for the first group is $O(\epsilon)$ in the $\epsilon$-neighborhood of $\theta_2^*$ and the loss function $V_2(\cdot)$ for the second group is $O(\epsilon)$ in the $\epsilon$-neighborhood of $\theta_1^*$. In addition, assume that the determinants of the Hessian at two local minimizers are the same. Then the ratio of the transition times between the two local minimizers for resampling is*

$$\frac{\mathbb{E}[\tau_{\theta_1^* \to \theta_2^*}^s]}{\mathbb{E}[\tau_{\theta_2^* \to \theta_1^*}^s]} = \exp\left(\frac{2}{\eta}\left(\frac{\delta V(\theta_1^*)}{a_1 \nabla V_1(\theta_1^*)\nabla V_1(\theta_1^*)^\top} - \frac{\delta V(\theta_2^*)}{a_2 \nabla V_2(\theta_2^*)\nabla V_2(\theta_2^*)^\top}\right)\right) + O(\sqrt{\epsilon})$$

*and the ratio for reweighting is*

$$\frac{\mathbb{E}[\tau_{\theta_1^* \to \theta_2^*}^w]}{\mathbb{E}[\tau_{\theta_2^* \to \theta_1^*}^w]} = \exp\left(\frac{2}{\eta}\left(\frac{f_1 \delta V(\theta_1^*)}{a_1^2 \nabla V_1(\theta_1^*)\nabla V_1(\theta_1^*)^\top} - \frac{f_2 \delta V(\theta_2^*)}{a_2^2 \nabla V_2(\theta_2^*)\nabla V_2(\theta_2^*)^\top}\right)\right) + O(\sqrt{\epsilon}).$$

See Appendix B for the proof. When the ratio is larger than 1, it means that $\theta_1^*$ is more stable than $\theta_2^*$. This result shows that for reweighting the relative stability of the two minimizers highly depends on the sampling proportions $(f_1, f_2)$. On the other hand, for resampling it is independent of $(f_1, f_2)$, which is precisely the desired result for a bias correction procedure. To see how the sampling proportions affect the behavior of reweighting, let us consider a simple case where $\theta_1^*$ is the global minimizer, $\nabla V_1(\theta_1^*)\nabla V_1(\theta_1^*)^\top = \nabla V_2(\theta_2^*)\nabla V_2(\theta_2^*)^\top$, $a_1 = \frac{1}{2} + \epsilon, a_2 = \frac{1}{2} - \epsilon$, and $f_1 \ll f_2$. This ensures that $\delta V(\theta_1^*) > \delta V(\theta_2^*)$ and the above ratio for resampling is larger than 1, which is the desired result. However, $f_1 \ll f_2$ implies that $\frac{f_1}{a_1^2} \ll 1, \frac{f_2}{a_2^2} \gg 1$, and the above ratio for reweighting is much smaller than 1, which means that the local minimizer $\theta_2^*$ is more stable than the global minimizer $\theta_1^*$.

## 5 EXPERIMENTS

This section examines the empirical performance of resampling and reweighting for problems from classification, regression, and reinforcement learning. As mentioned in the previous sections, the noise of stochastic gradient algorithms makes optimal learning rate selections much more restrictive for reweighting, when the data sampling is highly biased. In order to achieve good learning efficiency and reasonable performance in a neural network training, adaptive stochastic gradient methods such as Adam (Kingma & Ba, 2014) are applied in the first two experiments. We observe that resampling consistently outperforms reweighting with various sampling ratios when combined with these adaptive learning methods.

**Classification.** This experiment uses the Bank Marketing data set from (Moro et al., 2014) to predict if a client will subscribe a term deposit. After preprocessing, the provided data distribution over the variable "y" that indicates the subscription, is highly skewed: the ratio of "yes" and "no" is $f_1/f_2 = 4640/36548 \approx 1/7.88$. We assume that the underlying population distribution is $a_1/a_2 = 1$. We setup a 3-layer neural network with the binary cross-entropy loss function and train with the default Adam optimizer. The training and testing data set is obtained using train_test_split provided in sklearn[2]. The training takes 5 epochs with the batch-size equal to 100. The performance is compared among the baseline (i.e. trained without using either resampling or reweighting), resampling (oversample the minority group uses the *sample* with replacement), and reweighting. We run the experiments 10 times for each case, and then compute and plot results by averaging.

To estimate the performance, rather than using the classification accuracy that can be misleading for biased data, we use the metric that computes the area under the receiver operating characteristic curve (ROC-AUC) from the prediction scores. The ROC curves plots the true positive rate on the $y$-axis versus the false positive rate on the $x$-axis. As a result, a larger area under the curve indicates a better performance of a classifier. From both Table 1 and Figure 3, we see that the oversampling has the best performance compared to others. We choose oversampling rather than undersampling for the resampling method, because if we naively down sample the majority group, we throw away many information that could be useful for the prediction.

---

[2] https://scikit-learn.org/stable

|  | Baseline | Resampling | Reweighting |
|---|---|---|---|
| training loss | 0.3221 | **0.2602** | 0.2831 |
| roc_auc_score | 0.9277 | **0.9516** | 0.9312 |

Table 1: The loss takes the binary cross-entropy with a 3-layer neural network. We see that in average of 10 trials, the resampling method (oversampling) achieves the lowest training loss and highest ROC-AUC score over testing data among all tested cases.

**Nonlinear Regression.** This experiment uses the California Housing Prices data set[3] to predict the median house values. The target median house values, ranging from 15k to 500k, are distributed quite non-uniformly. We select subgroups with median house values $> 400k$ (1726 in total) and $< 200k$ (11767 in total) and combine them to make our dataset. In the preprocessing step, we drop the "ocean proximity" feature and randomly set $30\%$ of the data to be the test data. The remaining training data set with 8 features is fed into a 3-layer neural network. The population proportion of two subgroups is assumed to be $a_1/a_2 \approx 1$, while resampling and reweighting are tested with various sampling ratios $f_1/f_2$ near 11767/1726. Their performance of is compared also with the baseline. In each test, the mean squared error (MSE) is chosen as the loss function and Adam

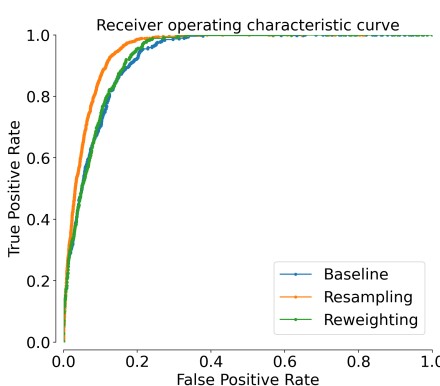

Figure 3: The ROC curve comparisons show that the resampling has the largest area under the curve.

is used as the optimizer in the model. The batch-size is 32 and the number of epochs is 400 for each case. As shown in Table 2, resampling significantly outperforms reweighting for all sampling ratios in terms of a lower averaged MSE, and its good stability is reflected in its lowest standard deviation for multiple runs.

| MSE | Baseline | RS | RW ($f_1/f_2 = 7$) | RW ($f_1/f_2 = 9$) | RW ($f_1/f_2 = 12$) |
|---|---|---|---|---|---|
| mean | 1.0386e+05 | **7.9679e+04** | 9.3567e+04 | 9.0436e+04 | 9.1949e+04 |
| std | 8.0371e+03 | **1.8620e+03** | 3.8044e+03 | 2.4692e+03 | 3.0341e+03 |

Table 2: Mean squared errors (MSE) for nonlinear regression problems. RS stands for resampling and RW for reweighting. The weights used in reweighting are $\frac{a_1}{f_1}$ and $\frac{a_2}{f_2}$, respectively. For each case, we run experiments for 10 times and compute the corresponding mean and standard deviation. Resampling (oversampling the minor group) achieves the lowest mean and standard deviation of MSE among all tested cases.

**Off-policy prediction.** In the off-policy prediction problem in reinforcement learning, the objective is to find the value function of policy $\pi$ using the trajectory $\{(a_t, s_t, s_{t+1})\}_{t=1}^{T}$ generated by a behavior policy $\mu$. To achieve this, the standard approach is to update the value function based on the behavior policy's temporal difference (TD) error $\delta(s_t) = R(s_t) + \gamma V(s_{t+1}) - V(s_t)$ with an importance weight $\mathbb{E}_\pi[\delta|s_t = s] = \sum_{a \in \mathbb{A}} \frac{\pi(a|s)}{\mu(a|s)} \mathbb{E}[\delta|s_t = s, a_t = a] \mu(a|s)$, where the summation is taken over the action space $\mathbb{A}$. The resulting reweighting TD learning for policy $\pi$ is

$$V_{t+1}(s_t) = V_t(s_t) + \eta \frac{\pi(a_t|s_t)}{\mu(a_t|s_t)} (R(s_t) + \gamma V_t(s_{t+1}) - V_t(s_t)),$$

where $\eta$ is the learning rate. This update rule is an example of reweighting. On the other hand, the expected TD error can also be written in the resampling form, $\mathbb{E}_\pi[\delta|s_t = s] = \sum_{a \in \mathbb{A}} \mathbb{E}[\delta|s_t =$

[3]https://www.kaggle.com/camnugent/california-housing-prices

$s, a_t = a]\pi(a|s) = \sum_{a \in \mathbb{A}} \sum_{j=1}^{\pi(a|s)N} \mathbb{E}[\delta^j | s_t = s, a_t = a]$, where $N$ is the total number of samples for $s_t = s$. This results to a resampling TD learning algorithm: at step $t$,

$$V_{t+1}(s_t) = V_t(s_t) + \eta(R(s_k) + \gamma V_t(s_{k+1}) - V_t(s_k)),$$

where $(a_k, s_k, s_{k+1})$ is randomly chosen from the data set $\{(a_j, s_j, s_{j+1})\}_{s_j = s_t}$ with probability $\pi(a_k | s_t)$.

Consider a simple example with discrete state space $\mathbb{S} = \{i\}_{i=0}^{n-1}$, action space $\mathbb{A} = \{\pm 1\}$, discount factor $\gamma = 0.9$ and transition dynamics $s_{t+1} = \mathrm{mod}(s_t + a_t, n)$, where the operator $\mod(m, n)$ gives the remainder of $m$ divided by $n$. Figure 4 shows the results of the off-policy TD learning by these two approaches, with the choice of $n = 32$ and $r(s) = 1 + \sin(2\pi s/n)$ and learning rate $\eta = 0.1$. The target policy is $\pi(a_i | s) = \frac{1}{2}$ while the behavior policy is $\mu(a_i | s) = \frac{1}{2} + ca_i$. The difference between the two policies becomes larger as the constant $c \in [0, 1/2]$ increases. From the previous analysis, if one group has much fewer samples as it should have, then the minimizer of the reweighting method is highly affected by the sampling bias. This is verified in the plots: as $c$ becomes larger, the performance of reweighting deteriorates, while resampling is rather stable and almost experiences no difference with the on-policy prediction in this example.

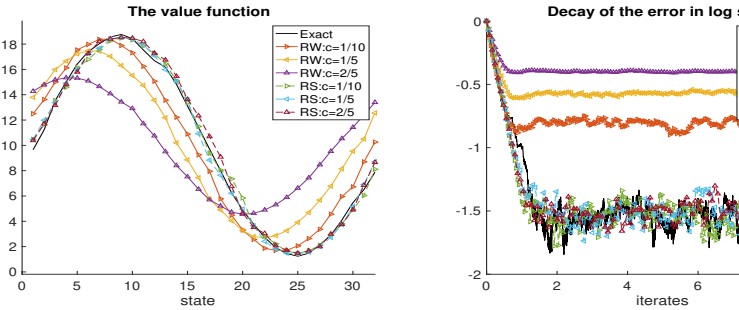

Figure 4: The left plot shows the approximate value function obtained by the two methods. The right plot is the evolution of the relative error $\log(\frac{e_t}{e_0})$, where the absolute error $e_t = \|V_t(s) - V^\pi(s)\|_2^2$. RW and RS in the upright corner represent reweighting and resampling, respectively. $c$ determines the behavior policy $\mu(a_i | s) = \frac{1}{2} + ca_i$. The value function is trained on a trajectory with length $10^5$ generated by the behavior policy. The value function obtained from resampling is fairly close to the exact value function, while the results of reweighting gets worse as the behavior policy gets further from the target policy.

## 6 DISCUSSIONS

This paper examines the different behaviors of reweighting and resampling for training on biasedly sampled data with the stochastic gradient descent. From both the dynamical stability and stochastic asymptotics viewpoints, we explain why resampling is numerically more stable and robust than reweighting. Based on this theoretical understanding, we advocate for considering data, model, and optimization as an integrated system, while addressing the bias.

An immediate direction for future work is to apply the analysis to more sophisticated stochastic training algorithms and understand their impact on resampling and reweighting. Another direction is to extend our analysis to unsupervised learning problems. For example, in the principal component analysis one computes the dominant eigenvectors of the covariance matrix of a data set. When the data set consists of multiple subgroups sampled with biases and a stochastic algorithm is applied to compute the eigenvectors, then an interesting question is how resampling or reweighting would affect the result.

## ACKNOWLEDGEMENTS

The work of L.Y. and Y.Z. is partially supported by the U.S. Department of Energy via Scientific Discovery through Advanced Computing (SciDAC) program and also by the National Science Foundation under award DMS-1818449. J.A. is supported by Joe Oliger Fellowship from Stanford University.

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

## A PROOFS IN SECTION 3

### A.1 PROOF OF LEMMA 1

*Proof.* In resampling, near $\theta = -1$ the gradient is $\theta + 1$ with probability $a_1$ and 0 with probability $a_2$. Let us denote the random gradient at each step by $W(\theta + 1)$, where $W$ is a Bernoulli random

variable with mean $\mathbb{E}(W) = a_1$ and variance $\mathbb{V}(W) = a_1 a_2$. At the learning rate $\eta$, the iteration can be written as

$$(\theta_{k+1} + 1) = (1 - \eta W)(\theta_k + 1).$$

The first and second moments of the iterates are

$$
\begin{aligned}
\mathbb{E}[\theta_k + 1] &= (1 - \eta a_1)^k (\theta_0 + 1), \\
\mathbb{E}[(\theta_k + 1)^2] &= ((1 - \eta a_1)^2 + \eta^2 a_1 a_2)^k (\theta_0 + 1)^2.
\end{aligned}
\tag{8}
$$

According to the definition of the stochastic stability, SGD is stable around $\theta = -1$ if the multiplicative factor of the second equation is bounded by 1, i.e.

$$(1 - \eta a_1)^2 + \eta^2 a_1 a_2 \leq 1. \tag{9}$$

Consider now the stability around $\theta = 1$, the iteration can be written as

$$(\theta_{k+1} - 1) = (1 - \eta W)(\theta_k - 1),$$

where $W$ is again a Bernoulli random variable with $\mathbb{E}(W) = a_2$ and $\mathbb{V}(W) = a_1 a_2$. The same computation shows that the second moment follows

$$\mathbb{E}[(\theta_k - 1)^2] = ((1 - \eta a_2)^2 + \eta^2 a_1 a_2)^k (\theta_0 - 1)^2.$$

Therefore, the condition for the SGD to be stable around $\theta = 1$ is

$$(1 - \eta a_2)^2 + \eta^2 a_1 a_2 \leq 1. \tag{10}$$

$\square$

## A.2 Proof of Lemma 2

*Proof.* In reweighting, near $\theta = -1$ the gradient is $\frac{a_1}{f_1}(\theta + 1)$ with probability $f_1$ and 0 with probability $f_2$. Let us denote the random gradient at each step by $W(\theta + 1)$, where $W$ is a Bernoulli random variable with $\mathbb{E}(W) = a_1$ and $\mathbb{V}(W) = f_1 f_2 \left(\frac{a_1}{f_1}\right)^2$. At the learning rate $\eta$, the iteration can be written as

$$(\theta_{k+1} + 1) \leftarrow (1 - \eta W)(\theta_k + 1).$$

Hence the second moments of the iterates are given by

$$\mathbb{E}[(\theta_k + 1)^2] = ((1 - \eta a_1)^2 + \eta^2 f_1 f_2 (a_1/f_1)^2)^k (\theta_0 + 1)^2.$$

Therefore, the condition for the SGD to be stable around $\theta = -1$ is

$$(1 - \eta a_1)^2 + \eta^2 f_1 f_2 \left(\frac{a_1}{f_1}\right)^2 \leq 1.$$

Consider now the stability around $\theta = 1$, the gradient is 0 with probability $f_1$ and $\frac{a_2}{f_2}(\theta - 1)$ with probability $f_2$. An analysis similar to the case $\theta = -1$ shows that the condition for the SGD to be stable around $\theta = 1$ is

$$(1 - \eta a_2)^2 + \eta^2 f_1 f_2 \left(\frac{a_2}{f_2}\right)^2 \leq 1.$$

$\square$

## B Proofs in Section 4

### B.1 Proof of Lemma 3

*Proof.* In resampling, with probability $a_1$ the gradients over the four intervals $(-\infty, -1)$, $(-1, 0)$, $(0, 1)$, and $(1, \infty)$ are $-1$, 1, $\epsilon$, and $\epsilon$. With probability $a_2$, they are $-\epsilon$, $-\epsilon$, $-1$, and 1 across these four intervals. The variances of the gradients are $a_1 a_2 (1 - \epsilon)^2$, $a_1 a_2 (1 + \epsilon)^2$, $a_1 a_2 (1 + \epsilon)^2$, $a_1 a_2 (1 - \epsilon)^2$, respectively, across the same intervals.

Since $\epsilon \ll 1$, the variance can be written as $a_1 a_2 + O(\epsilon)$ across all intervals. Then the SGD dynamics with learning rate $\eta$ can be approximated by

$$\theta_{k+1} \leftarrow \theta_k - \eta \left( V'(\theta_k) + \sqrt{a_1 a_2 + O(\epsilon)} W \right),$$

where $W \sim \mathcal{N}(0, 1)$ is a normal random variable. When $\eta$ is small, one can approximate the dynamics by a stochastic differential equation of form

$$d\Theta = -V'(\Theta) dt + \sqrt{\eta} \sqrt{a_1 a_2 + O(\epsilon)} dB$$

by identifying $\theta_k \approx \Theta(t = k\eta)$ (see Appendix C for details). The stationary distribution of this stochastic process is

$$p_s(\theta) = \frac{1}{Z} \exp \left( -\frac{2}{(a_1 a_2 + O(\epsilon))\eta} V(\theta) \right),$$

where $Z$ is a normalization constant. Plugging in $\theta = -1, 1$ results in

$$\frac{p_s(1)}{p_s(-1)} = \exp \left( -\frac{2}{(a_1 a_2 + O(\epsilon))\eta} (V(1) - V(-1)) \right) = \exp \left( -\frac{2}{a_1 a_2 \eta} (V(1) - V(-1)) + O(\epsilon) \right)$$

$$= \exp \left( -\frac{2}{a_1 a_2 \eta} (V(1) - V(-1)) \right) + O(\epsilon).$$

Under the assumption that $\epsilon \ll 1$, the last term is negligible. When $a_2 > a_1$, $V(\theta)$ is minimized at $\theta = 1$, which implies $-(V(1) - V(-1)) > 0$. Hence, this ratio is larger than 1. □

## B.2 PROOF OF LEMMA 4

*Proof.* In reweighting, with probability $f_1$ the gradients are $-\frac{a_1}{f_1}, \frac{a_1}{f_1}, \frac{a_1}{f_1}\epsilon$, and $\frac{a_1}{f_1}\epsilon$ over the four intervals $(-\infty, -1)$, $(-1, 0)$, $(0, 1)$, and $(1, \infty)$, respectively. With probability $f_2$, they are $-\frac{a_2}{f_2}\epsilon$, $-\frac{a_2}{f_2}\epsilon$, $-\frac{a_2}{f_2}$, and $\frac{a_2}{f_2}$. The variances of the gradients are $f_1 f_2 (\frac{a_1}{f_1} - \frac{a_2}{f_2}\epsilon)^2$, $f_1 f_2 (\frac{a_1}{f_1} + \frac{a_2}{f_2}\epsilon)^2$, $f_1 f_2 (\frac{a_1}{f_1}\epsilon + \frac{a_2}{f_2})^2$, and $f_1 f_2 (\frac{a_1}{f_1}\epsilon - \frac{a_2}{f_2})^2$, respectively, across the same intervals.

Since $\epsilon \ll 1$, the variance can be written as $f_1 f_2 \frac{a_1^2}{f_1^2} + O(\epsilon)$ for $\theta < 0$ and $f_1 f_2 \frac{a_2^2}{f_2^2} + O(\epsilon)$ for $\theta > 0$.

With $\theta_k \approx \Theta(k\eta)$, the approximate SDE for $\theta < 0$ is given by

$$d\Theta = -V'(\Theta) dt + \sqrt{\eta} \sqrt{f_1 f_2 \frac{a_1^2}{f_1^2} + O(\epsilon)} dB$$

while the one for $\theta > 0$ is

$$d\Theta = -V'(\Theta) dt + \sqrt{\eta} \sqrt{f_1 f_2 \frac{a_2^2}{f_2^2} + O(\epsilon)} dB$$

(see Appendix C for the SDE derivations). The stationary distributions for $\theta < 0$ and $\theta > 0$ are, respectively,

$$\frac{1}{Z_1} \exp \left( -\frac{2}{\left( f_1 f_2 \frac{a_1^2}{f_1^2} + O(\epsilon) \right) \eta} V(\theta) \right), \quad \frac{1}{Z_2} \exp \left( -\frac{2}{\left( f_1 f_2 \frac{a_2^2}{f_2^2} + O(\epsilon) \right) \eta} V(\theta) \right).$$

Plugging in $\theta = -1, 1$ results in

$$\frac{p_w(1)}{p_w(-1)} = \frac{Z_1}{Z_2} \exp \left( -\frac{2}{\left( f_1 f_2 \frac{a_2^2}{f_2^2} + O(\epsilon) \right) \eta} V(1) + \frac{2}{\left( f_1 f_2 \frac{a_1^2}{f_1^2} + O(\epsilon) \right) \eta} V(-1) \right) \tag{11}$$

$$= \frac{Z_1}{Z_2} \exp \left( -\frac{2 f_2/f_1}{a_2^2 \eta} V(1) + \frac{2 f_1/f_2}{a_1^2 \eta} V(-1) + O(\epsilon) \right).$$

The next step is to figure out the relationship between $Z_1$ and $Z_2$. Consider an SDE with non-smooth diffusion $d\Theta = -V'(\Theta)dt + \sigma dB$. The Kolmogorov equation for the stationary distribution is

$$0 = p_t = \left( V'(\theta)p + \left( \frac{\sigma^2}{2}p \right)_\theta \right)_\theta . \tag{12}$$

This suggests that $\sigma^2 p$ is continuous at the discontinuity $\theta = 0$. In our setting, since $V(0) = 0$, this simplifies to

$$\left( f_1 f_2 \frac{a_1^2}{f_1^2} + O(\epsilon) \right) \eta \cdot \frac{1}{Z_1} = \left( f_1 f_2 \frac{a_2^2}{f_2^2} + O(\epsilon) \right) \eta \cdot \frac{1}{Z_2}.$$

This simplifies to

$$\frac{Z_1}{Z_2} = \frac{f_1 f_2 \frac{a_1^2}{f_1^2} + O(\epsilon)}{f_1 f_2 \frac{a_2^2}{f_2^2} + O(\epsilon)} = \frac{a_1^2/f_1^2}{a_2^2/f_2^2} + O(\epsilon).$$

Inserting this into (11) results in

$$\frac{p_w(1)}{p_w(-1)} = \left( \frac{a_1^2/f_1^2}{a_2^2/f_2^2} + O(\epsilon) \right) \exp\left( -\frac{2f_2/f_1}{a_2^2 \eta}V(1) + \frac{2f_1/f_2}{a_1^2 \eta}V(-1) + O(\epsilon) \right)$$

$$= \frac{a_1^2/f_1^2}{a_2^2/f_2^2} \exp\left( -\frac{2f_2/f_1}{a_2^2 \eta}V(1) + \frac{2f_1/f_2}{a_1^2 \eta}V(-1) \right) + O(\epsilon).$$

By the assumption $\frac{f_2}{f_1} \leq \frac{a_2}{a_1}\sqrt{\frac{V(-1)}{V(1)}}$ and $V(1) < V(-1) < 0$, one has $\left( \frac{a_1}{a_2} \right)^2 \left( \frac{f_2}{f_1} \right)^2 \leq \frac{V(-1)}{V(1)} < 1$ and $-\frac{f_2/f_1}{a_2^2}V(1) \leq -\frac{f_1/f_2}{a_1^2}V(-1)$. Hence the above ratio is less than 1. $\qquad\square$

### B.3 EXTENDED RESULTS FOR 1-DIMENSION

Let us consider the population loss function $V(\theta) = a_1 V_1(\theta) + a_2 V_2(\theta)$ with,

$$V_1(\theta) = \begin{cases} h_1(\theta), & \theta \leq 0 \\ \epsilon\theta, & \theta > 0 \end{cases}, \quad V_2(\theta) = \begin{cases} -\epsilon\theta, & \theta \leq 0 \\ h_2(\theta), & \theta > 0 \end{cases},$$

where $h_1, h_2$ are strictly convex functions and continuously differentiable. We assume $V(\theta)$ has two local minimizers $\theta_1 < 0, \theta_2 > 0$ and the values are negative at local minima. Therefore, when $a_2 > a_1$, $\theta_2$ should be the global minimizer. In addition, we assume that the geometries of $h_1, h_2$ at two local minimizers are similar, i.e., $h_1(\theta_1) = h_2(\theta_2)$, $h_1'(\theta_1) = h_2'(\theta_2)$; if we set $g_i(\theta)$ to be the anti-derivative of $1/h_i'(\theta)$, then $g_1(\theta_1) = g_2(\theta_2)$. Moreover, we assume that the two disjoint convex functions are smooth at the disjoint point, i.e., $h_1'(0) = h_2'(0)$ and $g_1(0) = g_2(0)$. The following two lemmas extend Lemmas 3 and 4 to piecewise strictly convex function based on the above assumptions.

**Lemma 9.** *When $a_2 > a_1$, $V(\theta_2) < V(\theta_1)$. The stationary distribution for resampling satisfies the relationship*

$$\frac{p_s(\theta_2)}{p_s(\theta_1)} = \exp\left( \frac{2}{\eta}\left( \frac{1}{a_1} - \frac{1}{a_2} \right) \int_{\theta_1}^0 \frac{1}{h_1'(\theta)}d\theta \right) + O(\epsilon) > 1.$$

*Proof.* In resampling, with probability $a_1$ the gradients in the two intervals $(-\infty, 0), (0, \infty)$ are $h_1'(\theta), \epsilon$ respectively; with probability $a_2$ the gradients are $-\epsilon, h_2'(\theta)$ respectively. Therefore, the expectation of the gradients $\mu(\theta)$ is $a_1 h_1'(\theta) + O(\epsilon)$ in $(-\infty, 0)$ and $a_2 h_2'(\theta) + O(\epsilon)$ in $(0, \infty)$. The variance of the gradients $\sigma(\theta)$ is $a_1 a_2 h_1'(\theta)^2 + O(\epsilon)$ in $(-\infty, 0)$ and $a_1 a_2 h_2'(\theta)^2 + O(\epsilon)$ in $(0, \infty)$. The p.d.f $p_s(t, \theta)$ satisfies

$$\partial_t p_s = \partial_\theta\left( \mu p_s + \frac{\eta}{2}\partial_\theta(\sigma p_s) \right),$$

therefore, the stationary distribution $p_s(\theta)$ satisfies

$$\left( \mu + \frac{\eta}{2}\partial_\theta\sigma \right)p_s + \frac{\eta\sigma}{2}\partial_\theta p_s = 0, \quad \text{or equivalently,} \quad \left( \frac{2\mu}{\eta\sigma} + \frac{\partial_\theta\sigma}{\sigma} \right)p_s + \partial_\theta p_s = 0,$$

which implies $p_s(\theta) = \frac{1}{Z}e^{-F(\theta)}$ with normalization constant $Z = \int_{-\infty}^{\infty} e^{-F(\theta)}$, where

$$F(\theta) = \int_{-\infty}^{\theta} \frac{2\mu(\xi)}{\eta\sigma(\xi)} + \frac{\partial_\xi\sigma(\xi)}{\sigma(\xi)}d\xi = \begin{cases} F_1(\theta) - F_1(-\infty), & \theta \leq 0, \\ F_2(\theta) - F_2(0) + F_1(0) - F_1(-\infty), & \theta > 0. \end{cases} \quad (13)$$

By inserting $\mu, \sigma$ in different intervals, one has

$$\begin{cases} F_1(\theta) = \dfrac{2}{\eta a_2} \displaystyle\int \dfrac{1}{h_1'}d\theta + \log(a_1 a_2 (h_1')^2) + O(\epsilon); \\ F_2(\theta) = \dfrac{2}{\eta a_1} \displaystyle\int \dfrac{1}{h_2'}d\theta + \log(a_1 a_2 (h_2')^2) + O(\epsilon). \end{cases}$$

Hence, the ratio of the stationary probabiliy at two local minimizers $\theta_1 < 0, \theta_2 > 0$ is

$$\frac{p_s(\theta_1)}{p_s(\theta_2)} = \exp(-F(\theta_1) + F(\theta_2)) = \exp(-F_1(\theta_1) + F_2(\theta_2) + (F_1(0) - F_2(0)))$$

$$= \exp\left(-\frac{2}{\eta a_2}g_1(\theta_1) + \frac{2}{\eta a_1}g_2(\theta_2) + \log\left(\frac{h_2'(\theta_2)^2}{h_1'(\theta_1)^2}\right)\right) \cdot$$

$$\exp\left(\frac{2}{\eta a_2}g_1(0) - \frac{2}{\eta a_1}g_2(0) + \log\left(\frac{h_1'(0)^2}{h_2'(0)^2}\right)\right) + O(\epsilon),$$

where $g_i(\theta) = \int \frac{1}{h_i'}d\theta, i = 1, 2$. By the assumption that $g_1(\theta_1) = g_2(\theta_2)$ and $h_1'(\theta_1) = h_2'(\theta_2)$, $g_1(0) = g_2(0)$ and $h_1'(0) = h_2'(0)$ one has,

$$\frac{p_s(\theta_1)}{p_s(\theta_2)} = \exp\left(\frac{2}{\eta}(g_1(0) - g_1(\theta_1))\left(\frac{1}{a_2} - \frac{1}{a_1}\right)\right) + O(\epsilon),$$

Since $a_2 > a_1 > 0$, $\frac{1}{a_2} - \frac{1}{a_1} < 0$. Because of the strictly convexity of $h_1$, $h_1'(\theta) > 0$ in $(\theta_1, 0)$, therefore, one has $g_1(0) - g_1(\theta_1) = \int_{\theta_1}^{0} \frac{1}{h_1'(\theta)}d\theta > 0$. Therefore

$$\frac{p_s(\theta_1)}{p_s(\theta_2)} = \exp\left(\frac{2}{\eta}(g_1(0) - g_1(\theta_1))\left(\frac{1}{a_2} - \frac{1}{a_1}\right)\right) + O(\epsilon) < 1,$$

$\square$

**Lemma 10.** *When $a_2 > a_1$, $V(\theta_2) < V(\theta_1)$. Under the condition $\frac{f_1}{f_2} > \sqrt{\frac{a_1}{a_2}}$, the stationary distribution for resampling satisfies the relationship*

$$\frac{p_w(\theta_2)}{p_w(\theta_1)} = \exp\left(\frac{2}{\eta}\left(\frac{f_2}{f_1 a_2} - \frac{f_1}{f_2 a_1}\right)\int_{\theta_1}^{0}\frac{1}{h_1'(\theta)}d\theta\right) + O(\epsilon) < 1.$$

One sufficient condition such that $\frac{f_1}{f_2} > \sqrt{\frac{a_1}{a_2}}$ is when $f_1, f_2$ is significantly different from $a_1, a_2$ in the sense that $f_1 > f_2$ when the actually population proportion $a_1 < a_2$.

*Proof.* In reweighting, with probability $f_1$ the gradients over the two intervals $(-\infty, 0), (0, \infty)$ are $\frac{a_1}{f_1}h_1'(\theta), \frac{a_1}{f_1}\epsilon$ respectively; with probability $f_2$ the gradients are $-\frac{a_2}{f_2}\epsilon, \frac{a_2}{f_2}h_2'(\theta)$ respectively. Therefore, the expectation of the gradients $\mu(\theta)$ is $a_1 h_1'(\theta) + O(\epsilon)$ in $(-\infty, 0)$ and $a_2 h_2'(\theta) + O(\epsilon)$ in $(0, \infty)$. The variance of the gradients $\sigma(\theta)$ is $\frac{f_2}{f_1}a_1^2 h_1'(\theta)^2 + O(\epsilon)$ in $(-\infty, 0)$ and $\frac{f_1}{f_2}a_2^2 h_2'(\theta)^2 + O(\epsilon)$ in $(0, \infty)$. From the similar analysis as in Lemma 9, the stationary distribution is $p_w(\theta) = \frac{1}{Z}e^{-F(\theta)}$ with the same $F(\theta)$ defined in equation 13, but $F_1, F_2$ are defined as follows

$$\begin{cases} F_1(\theta) = \dfrac{2f_1}{\eta f_2 a_1} \displaystyle\int \dfrac{1}{h_1'}d\theta + \log\left(\dfrac{f_2 a_1^2}{f_1}(h_1')^2\right) + O(\epsilon); \\ F_2(\theta) = \dfrac{2f_2}{\eta f_1 a_2} \displaystyle\int \dfrac{1}{h_2'}d\theta + \log\left(\dfrac{f_1 a_2^2}{f_2}(h_2')^2\right) + O(\epsilon). \end{cases}$$

Hence, the ratio of the stationary probabiliy at two local minimizers $\theta_1 < 0, \theta_2 > 0$ is

$$\frac{p_w(\theta_1)}{p_w(\theta_2)} = \exp(-F_1(\theta_1) + F_2(\theta_2) + (F_1(0) - F_2(0)))$$

$$= \exp\left(-\frac{2f_1}{\eta f_2 a_1}g_1(\theta_1) + \frac{2f_2}{\eta f_1 a_2}g_2(\theta_2) + \log\left(\frac{f_1^2 a_2^2}{f_2^2 a_1^2}\frac{h_2'(\theta_2)^2}{h_1'(\theta_1)^2}\right)\right) \cdot$$

$$\exp\left(\frac{2f_1}{\eta f_2 a_1}g_1(0) - \frac{2f_2}{\eta f_1 a_2}g_2(0) + \log\left(\frac{f_2^2 a_1^2}{f_1^2 a_2^2}\frac{h_1'(0)^2}{h_2'(0)^2}\right)\right) + O(\epsilon),$$

where $g_i(\theta) = \int \frac{1}{f_i'}d\theta, i = 1, 2$. By the assumption that $g_1(\theta_1) = g_2(\theta_2)$ and $h_1'(\theta_1) = h_2'(\theta_2)$, $g_1(0) = g_2(0)$ and $h_1'(0) = h_2'(0)$ one has,

$$\frac{p_w(\theta_1)}{p_w(\theta_2)} = \exp\left(\frac{2}{\eta}(g_1(0) - g_1(\theta_1))\left(\frac{f_1}{f_2 a_1} - \frac{f_2}{f_1 a_2}\right)\right) + O(\epsilon).$$

Because of the strictly convexity of $h_1$, one has $g_1(0) - g_1(\theta_1) > 0$. By the assumption $\frac{f_1}{f_2} > \sqrt{\frac{a_1}{a_2}}$, then $\left(\frac{f_1}{f_2 a_1} - \frac{f_2}{f_1 a_2}\right) > 0$, which gives $\frac{p_s(\theta_1)}{p_s(\theta_2)} > 1$. $\qquad\square$

**Proof of Lemmas 5 and 6** We can further extend the results in 1D for a finite number of local minima as presented in Lemmas 5 and 6. In the same way as in the two local minima case, we assume that $h_i(\theta)$ has a similar geometry at the minimizers and $h_i(\theta), h_{i+1}(\theta)$ are smooth enough at the disjoint point $\theta_i$. In order to obtain the ratio of the stationary distribution at two arbitrary local minimizes, we take an additional assumption that $g_i(\theta_{i-1}) = g_i(\theta_i)$ for all $i$, where $g_i(\theta)$ is the anti-derivative of $1/h_i'(\theta)$. Intuitively, this assumption requires that each local minimum has an equal barrier on both sides. To be more specific, the assumptions we mentioned above are the following: at all the local minimizers, $h_i(\theta_i^*) = h_j(\theta_j^*) < 0, h_i'(\theta_i^*) = h_j'(\theta_j^*)$, let $g_i(\theta) = \int \frac{1}{h_i'(\theta)}d\theta$, then $g_i(\theta_i^*) = g_j(\theta_j^*)$ for any $i \neq j$; at all the disjoint points, $h_i'(\theta_i) = h_{i+1}(\theta_i), g_i(\theta_{i-1}) = g_i(\theta_i) = g_{i+1}(\theta_i)$ for all $i$. Lemmas 5 and 6 are under the above assumptions.

*Proof of Lemma 5.* For resampling, with probability $a_i$, the gradient is $h_i'(\theta)$ for $\theta \in (\theta_{i-1}, \theta_i)$, and $O(\epsilon)$ for $\theta \notin (\theta_{i-1}, \theta_i)$. Therefore, the expectation and variance in $(\theta_{i-1}, \theta_i)$ are $\mu = a_i h_i'(\theta) + O(\epsilon)$ and $\sigma = a_i(1 - a_i)h_i'(\theta)^2 + O(\epsilon)$. The stationary solution is

$$p_s(\theta) = \frac{1}{Z}e^{-F(\theta)}, \quad \text{with } F(\theta) = F_i(\theta) - F_i(\theta_{i-1}) + \sum_{j=1}^{i-1} F_j(\theta_j) - F_j(\theta_{j-1}), \text{ for } \theta \in (\theta_{i-1}, \theta_i),$$

where $Z = \int_{-\infty}^{\infty} e^{-F(\theta)}$ is a normalization constant and

$$F_i(\theta) = \frac{2}{\eta}\int \frac{1}{h_i'(\theta)}d\theta + \log\left(a_i(1 - a_i)h_i'(\theta)^2\right) + O(\epsilon).$$

Therefore, the ratio of the stationary probability at any two local minimizers $\theta_p^*, \theta_q^*$ is

$$\frac{p_s(\theta_p^*)}{p_s(\theta_q^*)} = \exp\left[-\left(F_p(\theta_p^*) - F_p(\theta_{p-1}) + \sum_{j=1}^{p-1} F_j(\theta_j) - F_j(\theta_{j-1})\right)\right.$$

$$\left. + \left(F_q(\theta_q^*) - F_q(\theta_{q-1}) + \sum_{j=1}^{q-1} F_j(\theta_j) - F_j(\theta_{j-1})\right)\right]$$

$$= \exp\left[-F_p(\theta_p^*) + F_q(\theta_q^*) + \sum_{j=p}^{q-1} F_j(\theta_j) - F_{j+1}(\theta_j)\right]$$

$$= \exp\left(-\frac{2}{\eta(1 - a_p)}g_p(\theta_p^*) + \frac{2}{\eta(1 - a_q)}g_q(\theta_q^*) + \log\left(\frac{a_q(1 - a_q)h_q'(\theta_q^*)^2}{a_q(1 - a_p)h_p'(\theta_p^*)^2}\right)\right) \cdot$$

$$\exp\left(\sum_{j=p}^{q-1} \frac{2}{\eta(1 - a_j)}g_j(\theta_j) - \frac{2}{\eta(1 - a_{j+1})}g_{j+1}(\theta_j^*) + \log\left(\frac{a_j(1 - a_j)h_j'(\theta_j)^2}{a_{j+1}(1 - a_{j+1})h_{j+1}'(\theta_j)^2}\right)\right) + O(\epsilon).$$

By the assumption that $g_p(\theta_p^*) = g_q(\theta_q^*), h_p'(\theta_p^*) = h_q'(\theta_q^*)$ and $g_i(\theta_{i-1}) = g_i(\theta_i) = g_{i+1}(\theta_i), h_i'(\theta_i) = h_{i+1}'(\theta_i)$ for all $i$, then the above ratio can be simplified to

$$\frac{p_s(\theta_p^*)}{p_s(\theta_q^*)} = \exp\left[\frac{2}{\eta}\left(g_p(\theta_p) - g_p(\theta_p^*)\right)\left(\frac{1}{1-a_p} - \frac{1}{1-a_q}\right)\right] + O(\epsilon) = \begin{cases} > 1, & \text{if } a_p > a_q; \\ < 1, & \text{if } a_p < a_q, \end{cases}$$

where the last inequality can be easily derived from that $g_p(\theta_p) - g_p(\theta_p^*) = \int_{\theta_p^*}^{\theta_p} \frac{1}{h_p'(\theta)} d\theta > 0$ because of the strictly convexity of $h_p$. $\square$

*Proof of Lemma 6.* For reweighting, with probability $f_i$, the gradient is $\frac{a_i}{f_i} h_i'(\theta)$ for $\theta \in (\theta_{i-1}, \theta_i)$, and $O(\epsilon)$ for $\theta \notin (\theta_{i-1}, \theta_i)$. Therefore, the expectation and variance in $(\theta_{i-1}, \theta_i)$ are $\mu = a_i h_i'(\theta) + O(\epsilon)$ and $\sigma = \frac{(1-f_i)a_i^2}{f_i} h_i'(\theta)^2 + O(\epsilon)$. The stationary solution

$$p_w(\theta) = \frac{1}{Z} e^{-F(\theta)}, \quad \text{with } F(\theta) = F_i(\theta) - F_i(\theta_{i-1}) + \sum_{j=1}^{i-1} F_j(\theta_j) - F_j(\theta_{j-1}), \text{ for } \theta \in (\theta_{i-1}, \theta_i),$$

where $Z = \int_{-\infty}^{\infty} e^{-F(\theta)}$ is a normalization constant and

$$F_i(\theta) = \frac{2f_i}{\eta a_i(1-f_i)} \int \frac{1}{h_i'(\theta)} d\theta + \log\left(\frac{(1-f_i)a_i^2}{f_i} h_i'(\theta)^2\right) + O(\epsilon)$$

Therefore, the ratio of the stationary probability at any two local minimizers $\theta_p^*, \theta_q^*$ is

$$\frac{p_w(\theta_p^*)}{p_w(\theta_q^*)} = \exp\left[-F_p(\theta_p^*) + F_q(\theta_q^*) + \sum_{j=p}^{q-1} F_j(\theta_j) - F_{j+1}(\theta_j)\right]$$

$$= \exp\left(-\frac{2f_p}{\eta a_p(1-f_p)} g_p(\theta_p^*) + \frac{2f_q}{\eta a_q(1-f_q)} g_q(\theta_q^*) + \log\left(\frac{f_p(1-f_q)a_q^2 h_q'(\theta_q^*)^2}{f_q(1-f_p)a_p^2 h_p'(\theta_p^*)^2}\right)\right) \cdot$$

$$\exp\left(\sum_{j=p}^{q-1} \frac{2}{\eta(1-a_j)} g_j(\theta_j) - \frac{2}{\eta(1-a_{j+1})} g_{j+1}(\theta_j^*) + \log\left(\frac{f_j(1-f_j)a_j^2 h_j'(\theta_j)^2}{f_{j+1}(1-f_{j+1})a_{j+1}^2 h_{j+1}'(\theta_j)^2}\right)\right) + O(\epsilon)$$

By the assumption that $g_p(\theta_p^*) = g_q(\theta_q^*), h_p'(\theta_p^*) = h_q'(\theta_q^*)$ and $g_i(\theta_{i-1}) = g_i(\theta_i) = g_{i+1}(\theta_i), h_i'(\theta_i) = h_{i+1}'(\theta_i)$ for all $i$, then the above ratio can be simplified to

$$\frac{p_w(\theta_p^*)}{p_w(\theta_q^*)} = \exp\left[\frac{2}{\eta}\left(g_p(\theta_p) - g_p(\theta_p^*)\right)\left(\frac{f_p}{a_p(1-f_p)} - \frac{f_q}{a_q(1-f_q)}\right)\right] + O(\epsilon).$$

$\square$

**Follow-up discussions of Lemma 5 and 6**

We first note that $\int_{\theta_p^*}^{\theta_p} \frac{1}{h_p'(\theta)} d\theta > 0$ due to the strictly convexity of $h_p$. Therefore, one can see from Lemma 5 that for resampling, the stationary solution always has the highest probability at the global minimizer. On the other hand, for the stationary solution of reweighting in Lemma 6, let us consider the case when $a_p > a_q$. In this case, $V(\theta_p^*) < V(\theta_q^*)$, therefore, one expects the above ratio larger than 1, which implies that $\frac{f_p}{a_p(1-f_p)} - \frac{f_q}{a_q(1-f_q)} > 0$. Note that if $f_p = a_p, f_q = a_q$, then this term is always larger than 0, but when $f_p, f_q$ are significantly different from $a_p, a_q$ in the sense that $f_p < f_q$ and $f_p < a_p, f_q > a_q$, then $\frac{f_p}{a_p(1-f_p)} - \frac{f_q}{a_q(1-f_q)} < 0$, which will lead to $\frac{p_s(\theta_p^*)}{p_s(\theta_q^*)} < 1$, i.e., higher probability of converging to $\theta_q^*$, which is not desirable. To sum up, Lemma 6 shows that for reweighting, the stationary solution won't have the highest probability at the global minimizer if the empirical proportion is significantly different fron the population proportion.

## B.4 PROOF OF LEMMA 8

*Proof.* By the variance of the gradients for resampling and reweighting in (5), and given that at the stationary point $\mathbb{E}[\nabla V(\theta_1^*)] = \mathbb{E}[\nabla V(\theta_2^*)] = 0$, one can omit the last term in the variance. In addition, since $\nabla V_1(\theta_2^*), \nabla V_2(\theta_1^*) = O(\epsilon) \ll \nabla V_1(\theta_1^*), \nabla V_2(\theta_2^*)$ by assumption, all the higher order terms are included in an $O(\sqrt{\epsilon})$ term. One can then derive Lemma 8 from Lemma 7. $\square$

## C  A JUSTIFICATION OF THE SDE APPROXIMATION

The stochastic differential equation approximation of SGD involving data-dependent covariance coefficient Gaussian noise was first introduced in (Li et al., 2017) and justified in the weak sense. Consider the SDE

$$d\Theta = b(\Theta)dt + \sigma(\Theta)dB. \tag{14}$$

The Euler-Maruyama discretization with time step $\eta$ results in

$$\Theta_{k+1} = \Theta_k + \eta b(\Theta_k) + \sqrt{\eta}\sigma(\Theta_k)Z_k, \ \ Z_k \sim \mathcal{N}(0,1), \ \ \Theta_0 = \theta_0. \tag{15}$$

In our case, $b(\cdot) = -V'(\cdot)$. When $b$ satisfies Lipschitz continuity and some technical smoothness conditions, according to (Li et al., 2017) for any function $g$ from a smooth class $\mathcal{M}$, there exists $C > 0$ and $\alpha > 0$ such that for all $k = 0, 1, 2, \cdots, N$,

$$|E[g(\Theta_{k\eta})] - E[g(\theta_k)]| \le C\eta^{\alpha}.$$

However, as the loss function considered in this paper has jump discontinuous in the first derivative, the classical approximation error results for SDE do not apply. In fact, the problem $V \notin C^1(\mathbb{R}^n)$ is a common issue in machine learning and deep neural networks, as many loss functions involves non-smooth activation functions such as ReLU and leaky ReLU. In our case, we need to justify the SDE approximation adopted in Section 3. It turns out that strong approximation error can be obtained if

- the noise coefficient $\sigma$ is Lipschitz continuous and non-degenerate, and
- the drift coefficient $b$ is piece-wise Lipschitz continuous, in the sense that $b$ has finitely many discontinuity points $-\infty = \xi_0 < \xi_1 < \cdots < \xi_m < \xi_{m+1} = \infty$ and in each interval $(\xi_{i-1}, \xi_i)$, $b$ is Lipschitz continuous.

Under these conditions, the following approximation result holds: for all $k = 0, 1, 2, \cdots, N$, there exists $C > 0$ such that

$$E[|\Theta_{k\eta} - \theta_k|] \le C\sqrt{\eta}. \tag{16}$$

Here $\Theta_{k\eta}$ is the solution to SDE at time $k\eta$. The proof strategy closely follows from (Müller-Gronbach et al., 2020). The key is to construct a bijective mapping $G : \mathbb{R} \to \mathbb{R}$ that transforms (14) to SDE with Lipschitz continuous coefficients. With such a bijection $G$, one can define a stochastic process $Z : [0, T] \times \Omega \to \mathbb{R}$ by $Z_t = G(\Theta_t)$ and the transformed SDE is

$$dZ_t = \tilde{b}(Z_t)dt + \tilde{\sigma}dB_t, \ \ t \in [0, T], \ \ Z_0 = G(\Theta_0), \tag{17}$$

$$\text{with } \tilde{b} = (G' \cdot b + \frac{1}{2}G'' \cdot \sigma^2) \circ G^{-1} \text{ and } \tilde{\sigma} = (G' \cdot \sigma) \circ G^{-1}. \tag{18}$$

As the SGD updates can essentially be viewed as data from the Euler-Maruyama scheme, considering $Z_k$ as updates from Euler-Maruyama scheme leads to

$$\mathbb{E}[|\Theta_{k\eta} - \theta_k|] \le c_1 \mathbb{E}[|Z_{k\eta} - G \circ \theta_k|] = c_1 \mathbb{E}[|Z_{k\eta} - Z_k + Z_k - G \circ \theta_k|]$$
$$\le c_2\sqrt{\eta} + c_1 \mathbb{E}[|Z_k - G \circ \theta_k|].$$

To control the second item, we introduce

$$\theta_t := \theta_k + b(\theta_k)(t - k\eta) + \sqrt{t - k\eta}\sigma(\theta_k)Z_k,$$

where $t \in [0, k\eta]$. Then as shown in (Müller-Gronbach et al., 2020),

$$\mathbb{E}[|Z_k - G \circ \theta_k|] \le c\sqrt{\eta} + c\mathbb{E}\left[\left|\int_0^{k\eta} 1_B(\theta_t, \theta_k)dt\right|\right],$$

with $B$ being the set of pairs $(y_1, y_2) \in \mathbb{R}^2$ where the joint Lipschitz estimate $|b(y_1) - b(y_2)|$ does not apply due to at least one discontinuity. In (Müller-Gronbach et al., 2020), it is estimated by

$$\mathbb{E}\left[\left|\int_0^{k\eta} 1_B(\theta_t, \theta_k)dt\right|\right] \le c\sqrt{\eta},$$

which leads us to (16).

# D NUMERICAL COMPARISONS WITH DIFFERENT LEARNING RATES

In this section, we present extensive numerical results to show the effect of learning rates in our toy examples. The Figure 5 corresponds to the example in Section 3, and Figure 6 corresponds to the example in Section 4.

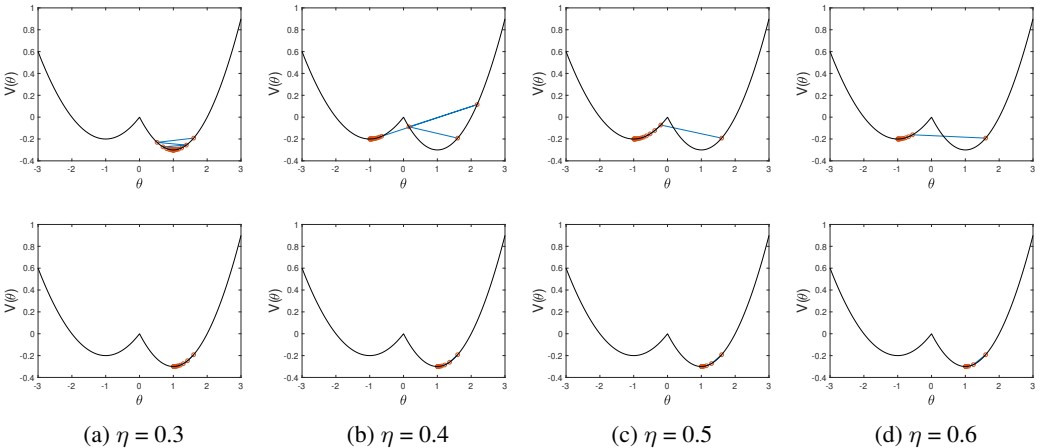

(a) $\eta = 0.3$      (b) $\eta = 0.4$      (c) $\eta = 0.5$      (d) $\eta = 0.6$

Figure 5: A comparison of reweighting (upper row) and resampling (lower row) with $a_1/a_2 = 0.4/0.6$ and $f_1/f_2 = 0.9/0.1$ at various learning rates $\eta$. All experiments start at $\theta_0 = 1.6$. We can see that unless the learning rate $\eta < 0.4$, resampling is more stable near the minimizer $\theta = 1$.

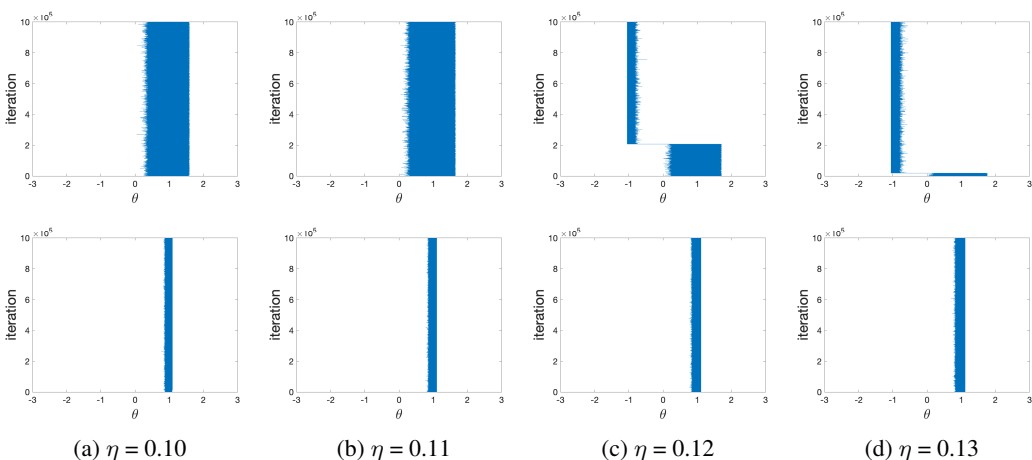

(a) $\eta = 0.10$      (b) $\eta = 0.11$      (c) $\eta = 0.12$      (d) $\eta = 0.13$

Figure 6: A comparison of reweighting (upper row) and resampling (lower row) with $a_1/a_2 = 0.4/0.6$, $f_1/f_2 = 0.9/0.1$ and $\epsilon = 0.1$ at various learning rates $\eta$. All experiments start at $\theta_0 = 0.9$. We can see that unless the learning rate $\eta < 0.12$, resampling is more reliable in the sense that its trajectory stays around the desired minimizer.

