# OpenReview forum: "Why resampling outperforms reweighting for correcting sampling bias with stochastic gradients"
_ICLR.cc/2021/Conference — ICLR 2021 Poster_

### Official Review · AnonReviewer3 · 2020-10-25
**Paper provides explanation why resampling is better than reweighing for addressing bias problem in supervised learning**

**Rating:** 7
**Confidence:** 3

**Review:**

Biased datasets are ubiquitous and therefore addressing bias in datasets is a relevant issue for building good ML applications and products. The paper makes a significant contribution by providing a theoretical explanation to the observation that resampling is generally more effective than reweighing as a debiasing tool. This paper does not introduce a novel way to address bias and therefore its originality and impact is limited. It does however provide good proof that the resampling approach is more efficient than reweighing, which will help ML developers to make a more informed choice between the two approaches. The ubiquity of the sampling problem, makes the impact of this work significant despite not being very original. In addition, to the results, the approach used in the paper, namely, the use of numerical methods borrowed from stability analysis of dynamical systems and stochastic differential equations continues an interesting line of research that is helping to understand optimization processes and how to use them efficiently to train unsupervised algorithms. The theoretical results look solid and I cannot see any issue with them to the best of my knowledge. The authors provide and array of numerical results which convincingly demonstrate their theory. The manuscript could be improved by exploring the limitations of their approach, but I do not see this is needed for the paper to be accepted.

---

> ### Author Response · Authors · 2020-11-18
> **Response to AnonReviewer3**
>
> Thanks for the comments. We would like to carry out some novel methods to analyze the bias, but the main purpose of our paper is to make comparisons between two classical sample bias correcting methods. We have included more discussions in the end to specify the direction of future work.

---

### Official Review · AnonReviewer4 · 2020-10-28
**Why resampling outperforms reweighting for correcting sampling bias**

**Rating:** 5
**Confidence:** 4

**Review:**


Summary:
When training data comes from a sampling distribution that is different from the target test distribution, there are two commonly-used machine learning techniques to correct the distribution difference --- re-sampling and re-weighting. This paper investigates why re-sampling outperforms re-weighting when using stochastic gradient descent for optimization. The paper provides two explanations.
(1) Resampling with SGD is more stable. By stability, the authors mean that the expected L2 distance between the parameter during optimization and the true parameter is small.
(2) Reweighting with SGD can converge to a worse local minimum with larger probability.


They also conduct experiments to show that re-sampling outperforms re-weighting.

Strengths:

1. The problem to compare re-sampling and re-weighting during optimization with stochastic gradient descent is very interesting and important. Machine learning from biased data (e.g. selection bias) is very prevalent and stochastic gradient descent is a popular optimizer.

2. The paper is clearly written and easy to follow.

3. The stability analysis and SDE analysis provide two alternatives to explain the difference between re-sampling and re-weighting.

Weakness

1 The theoretical analysis does not provide strong evidence that re-sampling is better than re-weighting.

1.1 It would be great if the authors could explain more why Lemma 1 and Lemma 2 show that resampling is better. To me, it just shows that, if we want to achieve stability, we have to use smaller learning rate for reweighting. It makes no sense to me to compare resampling and reweighting with a fixed learning rate.

1.2 SDE analysis shows that re-weighting sometimes tends to converge to a worse local minimum. But it did not sate re-sampling always tends to converge to a good local minimum. It would be great if authors explain when re-weighting and re-sampling tend to converge to good and bad local minimum.




2. The experiments details are not clear and I did not see how these experiments reflect the theoretical analysis.


2.1 It would be great to mention whether the results are performed on the training set. I assume the results presented are for the training set since this paper investigates how the optimization with SGD works instead of focusing on generalization.

2.2 It would be great to report the final training objective. Since the main theoretical result of the paper is that optimization with SGD using re-sampling tends to be more stable around a better local minimum.

2.3 In the classification experiments, the ROC-AUC of reweighting method is only around 0.53, that is just 0.03 better than random guess. A large neural network typically can overfit a classification dataset with just two classes. I am wondering why the ROC-AUC is so low. It would be great if the authors could provide the experiment details, e.g. the learning rate.

2.4 In the nonlinear regression experiment, logistic regression is used. Logistic regression objective is a convex objective. There is no local minimum. So the experiments do not validate the SDE analysis. The authors claimed that they do this experiment to show that ``performance of SGD deteriorates when reweighting is used''. I did not see any previous analysis indicating this. It would be great if the authors could explain more about why they conduct this experiment and how this relates to their theoretical analysis.






Additional feedback

In the abstract ''reweighting outperforms resampling'' -> ''resampling outperforms reweighting''

Why in equation (3), there is an approximate equal? It seems to me that it is just equal. Same for equation (4).

learning rate \eta first appears in Lemma 1 without explanation. Also, Lemma 1 uses C= 1 which is not explained in the main paper.


In SDE analysis: The statement "all iterates will stay in this region'' confuses me. If the learning rate is large enough, I think it can get out of the the region. Say when \hat{\theta}_t is at -2 and we encounter an example from V_1, then the gradient is positive. If the learning rate is large enough, it can get to (0,\infty).

---

> ### Author Response · Authors · 2020-11-18
> **Response to AnonReviewer4**
>
> We would like to thank the reviewer for very careful evaluations and constructive suggestions. Here are our responses.
>
> **1.1**: We don't claim that resampling is always better than reweighting, our claim is that without knowing the optimal learning rates a priori, resampling tends to perform better than reweighting in stochastic gradient algorithms with high probability. It is true that when learning rates are sufficiently small, reweighting and resampling can both perform well, and we added numerical evidence of that in Appendix D in the revised manuscript. However, in practice if the problem we deal with is very stiff, we typically don't choose learning rates to be close to zero for efficiency consideration. In that case, resampling takes an advantage.
>
> **1.2**: Since we are studying a stochastic optimization algorithm, there is no deterministic result. We can only claim results with high probability rather than say "always". Our Lemmas 3 and 4 imply that resampling is more likely to converge to the good minimum than reweighting. Lemma 6 implies that the resampling method is expected to stay in the good minimizer more longer than the bad one, while the reweighting method is expected to do the opposite when sampled proportion $f_i$ is significantly different from the population proportion $a_i$.
>
> **2.1**:
> - For the first numerical experiment, we have both training and testing data sets available from the Porto Seguro's safe driver prediction dataset. We train the model by applying a 10-fold cross validation on the training dataset, and then evaluate on the testing dataset.
> - For the second numerical experiment (now replaced by a new regression problem), we only have training dataset avaiable. In this case, we apply a random train\_test\_split and use 30% of the training data as test data to evaluate the performance.
> - For the off-policy prediction, we use the TD update based on a trajectory with length $10^5$ generated by the behavior policy.
>
> **2.2**:
> - The first experiment uses binary cross-entropy as the loss, the Adam optimizer, and accuracy as the metric in the neural network model. We train with $20$ epochs and a batch size $1000$ for all the methods. In average of the cross-validation, the loss output from reweighting is around $0.099$, while the loss outputs from resampling and untreated imbalanced case are around $0.67$.
> - The second experiment uses the mean squared error (MSE) as the loss and the Adam optimizer in the neural network model. We have reported the MSE statistics for all methods in Table 1 in the revised paper.
> - The off-policy prediction, although TD method do not have an explicit loss function, the update rule is similar to SGD: $\theta_{t+1} = \theta_t + \xi(\theta_t)$, where $\xi$ is an unbiased estimate for the TD error $\delta^s_\pi(\theta) = \sum_{i=1}^m\delta^s_i(\theta)\pi(a_i|s)$, where $\delta^s_i(\theta) = R(s) + \gamma V_\theta(s_{t+1}|s_t = s,a_t =a_i) - V_\theta(s_t)$. So in the off-policy setting, the population gradient is $\delta^s_\pi(\theta) = \sum_{i=1}^{m} \delta^s_i(\theta) \pi(a_i|s)$, while the empirical gradient is $\delta^s_\mu(\theta) = \sum_{i=1}^{m} \delta^s_i(\theta) \mu(a_i|s)$. The population proportion for group $i$ is $\pi(a_i|s)$, which is equivalent to $a_i$ in the setting of Section 2; while the empirical proportion of samplings for group $i$ is $\mu(a_i|s)$, which is equivalent to $f_i$ in the setting of Section 2.
> **2.3** Our guess is that when experimenting with various ratios $f_1/f_2$, if $f_1, f_2$ are far from the true population proportion, then reweighting can possibly perform even worse than the baseline. As we don't know the true population proportion but assume that $a_1/a_2 = 26.4/1$ according to the training data set we have, it is likely that we didn't weight each sample by a correct weight due to the missing information. When the associated weight $a_i/f_i$ is far from the appropriate weight, applying reweighting is almost equivalent to random guess. However, as our numerical results under different ratios $f_1/f_2$ show, reweighting's performance can improve when $f_1/f_2$ is adjusted to the true proportion, but still for a wide range of $f_1/f_2$ resampling performs better.
>
> **2.4**: Thanks for pointing this out. We have replaced the second experiment by another nonlinear regression problem trained by the neural network. Please check details in the revised paper that we uploaded.
>
> *Response to additional feedback*
>
> Thanks for pointing out typos, we have corrected them in the revised manuscript.
> - It should be  "resampling outperforms reweighting".
> - The approximation should be exactly equality.
> - We have clarified the learning rate and $0< C\leq 1$ (for contracting over $k$ iterations) in Lemma 1.
> - In SDE analysis, the SDE approximates SGD for sufficiently small learning rates. Therefore, assuming "all iterates will stay in this region" is reasonable.

---

> > ### Comment · AnonReviewer4 · 2020-11-23
> > **Thanks for your response**
> >
> > Thanks for your response and the additional non-linear experiments are more appropriate to show the theoretical results. But I still have the following concerns:
> >
> > 1.1 ''resampling tends to perform better than reweighting in stochastic gradient algorithms with high probability''. I do not get why Lemma 1 and Lemma 2 show this. Lemma 1 and Lemma 2 show that we should use different learning rates for resampling and reweighting. Numerical evidence in Appendix D uses the same small range of learning rates for both methods which is not what Lemma 1 and Lemma 2 tell us to do. How about we see different range of learning rates for both methods? Think about the objective that is 100 times the resampling objective, the learning rate should be small for this objective to work and the variance is 10000 times that of the reweighting objective. So smaller optimal learning rate or smaller gradient variance does not mean ''perform better with high probability''.
> >
> > 1.2 I might fail to explain my concern clearly. I was referring to Lemma 5 and Lemma 6 and  I did not mean we should get some results that show A is "always" better than B. I am curious when each method more likely converge to good local minimum and when each method more likely converge to bad local minimum. The paper only gives one simple case where reweighting converges to bad local minimum. As the authors mentioned, it is not always the case that reweighting is better than resampling. Does resampling more likely converge to a good local minimum no matter that setting it is？ Does reweighting more likely converge to a bad one no matter what setting it is? If not, when would they succeed and fail? These explanations could more help the readers understand the whole picture of the theoretical results.
> >
> > 2.1 I am confused why we should report results on the test set. The theoretical results compare the training performance and seem do not care about generalization. I do not see how results on test set could explain any theoretical results.
> >
> > 2.2.The training loss of reweighting is smaller. How does this validate the theoretical claims?
> >
> > 2.3. We can better know why it performs so bad on the test set if we look at the training performance.
> >
> > Due to the above concerns about theoretical and empirical results, I could not change my current evaluation.

---

> > > ### Author Response · Authors · 2020-11-24
> > > **Response to AnonReviewer4's further comments**
> > >
> > > We would like to thank the reviewer for further comments. Here are our responses.
> > >
> > > **1.1:** The quoted statement is more appropriate for the SDE analysis, while Lemmas 1 and 2 state the advantage of resampling over reweighting from the stability viewpoint (in the expectation sense). Lemmas 1 and 2 tell that resampling is more stochastically stable for a wider range of learning rates. Our numerical results in Appendix D do reflect what Lemma 1 and 2 indicate: in Figure 5, when we start from $\eta=0.3$, both methods perform well, but as we increase the learning rate $\eta$, reweighting becomes unstable near the local minimizer $\theta=1$ while resampling trajectories still stay around $\theta=1$.
> > >
> > > In practice, we often require the learning rates not to be too small in order for the SGD to sufficiently explore the loss landscape, so that it can converge to a global rather than a local minimum. With Lemmas 1 and 2's stochastic stability conditions in mind, it is, therefore, better to choose resampling over reweighting if one wants both stability and exploration efficiency.
> > >
> > > **1.2:** For Lemma 6, since the case of $a_1<a_2$ is similar to the case of $a_1>a_2$, so we only discuss the case that $a_1>a_2$ here.
> > > Let $G_1 = \frac{2}{\eta}\frac{\delta V(\theta_1^*)}{\nabla V_1(\theta_1^*) \nabla V_1(\theta_1^*)^\top} , \quad G_2 = \frac{2}{\eta}\frac{\delta V(\theta_2^*)}{\nabla V_2(\theta_2^*) \nabla  V_2(\theta_2^*)^\top}.$
> > > As long as $a_1>a_2, f_1<f_2$,  then the following always holds,
> > > $$\frac{G_1}{a_1} - \frac{G_2}{a_2} > \frac{f_1 G_1}{a_1^2} - \frac{f_2 G_2}{a_2^2}, \quad \text{or equivalently,}\quad a_1^2(a_1 - f_1)G_1 > a_1^2(f_1 - a_1)G_2, $$
> > > where the second inequality is obtained by replacing $a_2 =1-a_1, f_2 = 1-f_1$ and the fact that $a_1 - f_1 > 0$ and $G_1, G_2 > 0$. Therefore, the following relationship,
> > > $\frac{\mathbb{E}[\tau_{\theta_1\to\theta_2}^s]}{\mathbb{E}[\tau^s_{\theta_2\to\theta_1}]} > \frac{\mathbb{E}[\tau_{\theta_1\to\theta_2}^w]}{\mathbb{E}[\tau_{\theta_2\to\theta_1}^w]}$
> > > always holds for $\epsilon$ small enough, which implies that resampling always performs more stable at the global minimum than reweighting.
> > >
> > > Although when $a_1>a_2$, $f_1 \gg f_2$, one could have the opposite, we cannot know $a_1,a_2$ a priori, so it is still dangerous to use resampling in SGD. However, the ratio of resampling in Lemma 6 is independent of $(f_1,f_2)$, which is precisely the desired result for a bias correction procedure.
> > > One could also see it in Lemmas 3 and 4. If the condition on $f_2/f_1$ in Lemma 4 is not satisfied, it is possible that reweighting could have a higher probability on the global minimum, and reweighting could even perform more stable on the global minimum than resampling.   However, in general, since $a_1,a_2,V(-1),V(1)$ is not known a priori, so it is always safe to use resampling in SGD because it always has a higher probability on global minimum no matter how the sampling proportion changes.
> > >
> > > **2:** When we apply the 10-fold cross-validation of the training data set, we also evaluate the performance on the local test data set generated from StratifiedKFold(X\_train, y\_train). Therefore, we don't have the generalization result but only compare the training performance. We are sorry for mistakenly stating the test data set in the paper. We have corrected the wording.

---

> > > > ### Comment · AnonReviewer4 · 2020-11-25
> > > > **Main concern about experiment results**
> > > >
> > > > Thanks for your timely response! I still have the following concerns.
> > > >
> > > >
> > > > 1.1 I do not think wider learning rate range is better. Learning rate is relative to the magnitude of the objective. We can just divide the reweighting objective by a larger number, then its stable learning rate range becomes larger. Or we can multiply the resampling objective by a large number, then its stable learning rate range becomes smaller. It really makes no sense to compare two objectives with the same learning rate.
> > > >
> > > > 1.2 Thanks for the explanation. It would be great to discuss this in the main paper.
> > > >
> > > > 2. My main concern is about experiments. The theoretical analysis is all about training objective, whether the training objective is more stable in a good local minimum. I would expect the experiments to show the training objective of resampling is better (i.e. the loss smaller). However, in the experiments in the paper, the authors report some other performance measures that are different training objective, which do not directly validate their theoretical analysis.
> > > >
> > > > From the author response, they mentioned that the training objective of reweighting is smaller than resampling, which is in contradiction to what their theoretical analysis suggests.
> > > >
> > > > I am still confused which data the authors trained on and which data the authors tested on. From the author response, they mentioned that "For the first numerical experiment,...., applying a 10-fold cross validation on the training dataset, and then evaluate on the testing dataset" and "we apply a random train_test_split and use 30% of the training data as test data to evaluate the performance". It seems to me that the objective that optimize is calculated from one set of data and the evaluation is conducted on another set of data. What I expect is the comparison of training objectives of reweighting and resampling since the theorems only analyzed whether resampling or reweighting is stable around a good local minimum of the training objective.

---

> > > > > ### Author Response · Authors · 2020-11-25
> > > > > **Reply to your concerns**
> > > > >
> > > > > **1.1** We're sorry that we did not make it clear. In fact, the objectives for reweighing and resampling are designed to make the loss function to be (statistically) the same. Please see the eqns (3) and (4) of the paper.
> > > > >
> > > > > **2** Sorry, that was a typo. The loss of reweighing is larger than the loss of resampling. We have rerun the numerical experiments (the experiment setup has changed a bit during the revision period). Currently, the averaged loss for resampling is 0.72, while the averaged loss for reweighting is 1.9.
> > > > >
> > > > >
> > > > > “For the first numerical experiment,…., applying a 10-fold cross validation on the training dataset, and then evaluate on the testing dataset” was referring to the first experiment (classification). Here the data is split into 10 groups. In each run, we pick one group to train and pick one group to test. This is iterated over multiple pairs picked by StratifiedKFold.
> > > > >
> > > > > “we apply a random train_test_split and use 30% of the training data as test data to evaluate the performance” was referring to the second experiment (regression). Here the data is split into two groups (70% vs 30%). The 70% group is used for training and the test is on the 30% group.

---

> > > > > > ### Comment · AnonReviewer4 · 2020-11-25
> > > > > > **I do not get why we do not report the training objective that is theoretically analyzed.**
> > > > > >
> > > > > > Thanks for the clarification.
> > > > > >
> > > > > > 1.1 I guess you are saying the gradient in each step of the SGD  is in expectation the same. It would be great to mention this in the paper.
> > > > > >
> > > > > > 2. My main concern is about the experiments. The theorems analyzed the stability of training objective. The experiments report the performance measure that is different from the training objective. First, the performance is calculated on another dataset instead of the dataset they trained on. Second, the performance measure is different from the training performance measure. As a result, the empirical results do not validate the theorems. It would be great to report the training objectives in the paper.
> > > > > >
> > > > > > I agree that the performance on a test set might be interesting. But only reporting training objective can we validate the theoretical analysis.

---

> > > > > > > ### Author Response · Authors · 2020-11-25
> > > > > > > **Address to your concerns**
> > > > > > >
> > > > > > > **1.1**  Thank you. We included a sentence about this (the first blue sentence after equation 4). Thanks!
> > > > > > >
> > > > > > > **2**  We agree with your concern. In the training and testing data sets, the $a_1$ and $a_2$ proportions are the same (up to minor fluctuation due to data generation). Our original motivation is to report the testing (generalization) error in order to avoid the potential overfitting problem of the neural network.
> > > > > > >
> > > > > > > We now have modified the manuscript to include the training error (please see the first paragraph at the top of page 8).

---

### Official Review · AnonReviewer1 · 2020-10-28

**Rating:** 6
**Confidence:** 3

**Review:**

Summary:

This paper delves into a stability analysis of reweighting and resampling for overcoming imbalanced data in supervised learning. Reweighting employs the use of importance ratios to modify a samples weight to the training in turn changing the effective distribution. There are several resampling procedures which all have a similar effect in the analysis, and the authors consider several algorithms for resampling in their experiments.  The reweighting approach, while convenient, leads to poorer stability under simplifying assumptions. While this is interesting in its own right, they show that under certain distributions of the data reweighting will actually not converge to the optimal minima, while resampling will. This is motivated by a large collection of work developing resampling methods for imbalanced data, which all come to similar conclusions (i.e. that following a resampling procedure outperforms a reweighting procedure in many, but not all, settings). They follow up with a SDE analysis in another toy problem, which they then extend to more realistic assumptions.


Thoughts:

This paper is a good start at trying to understand the behavior of resampling and reweighting in the wild. It is also well written, and contains some interesting discussion on their analysis. The proofs also seem straightforward for the most part and well explained. Unfortunately, I have a few concerns (see below) which have lowered my score. I'm recommending reject at this time, but would be happy to increase my score as we go through the rebuttal phase.


Concerns/suggestions/questions:

1. My first concern has to do with how general we can expect the analysis of these problems to be. I understand that the analysis for larger problems is often harder if not impossible given our current theory, but if we simplify too much the results aren't indicative of the larger picture. Some assumptions which I have some issues with
   - Our problem can be decomposed into several discrete loss functions (i.e. $V_1(\theta)$ and $V_2(\theta)$). I can see how this works for classification where each loss function is for each label, but when moving to regression and RL prediction, this becomes less clear. For the case of learning a value function off-policy, it feels more like the importance weights are shifting the space, rather than reweighting a discrete set of loss functions.
   - For the general results you still only consider the case where there are two minimizers. (Q1-1) Are you only considering the case where the weights only shift within some local ball and only encounters these minima? (Q1-2) Can this be guaranteed in the larger problem you motivate (i.e. using NNs)? Maybe this is close to Lazy training, or results when the NN is overparameterized and the weights don't shift much. (Q1-3) Could these results be extended to a finite number of minimizers within a ball? This seems more reasonable.

These are the main assumptions I'm concerned with. They both are hinting at the larger question of (Q1-4) "how reasonable are these assumptions to the larger motivating problem?" I think it would make the paper much better if there was some examples that were less toy-like.


2. The results in the toy examples for the stability and SDE analysis are really nice. But I'm a bit concerned with the conclusions drawn. Both analysis results in bounds on a notion of "stability" which includes the learning rate as a factor. When testing the result empirically, only a single learning rate is used for both the resampling and reweighting. This gives a facade of a results that reweighting is always unstable, which is not true. When selecting a stepsize in the correct range reweighting should indeed be stable. I think what the authors are wanting to say is that selecting learning rates will be much easier for resampling as they do not rely on the sampling proportions. I agree with the sentiment, but I think this needs to be made more clear. One way is to give effective ranges for the learning rate in which we expect the two approaches to be stable, and observe that in practice we can't always know $(f_1, f_2)$ so selecting in the prober range is an empirical exercise.  You can also show this through uniformly selecting learning rates in the range $(0,1)$ and showing how many result in a stable system around the global minimizer. With these experiments, I would also suggest starting both procedures at the same initial weights and indicate through a color gradient or arrows the temporal direction of the training (right now it is quite ambiguous).

3. Experiments:

   I'm q bit concerned with the lack of details provided for the experimental design. I understand that in theory papers empirical results are not the focus, but the goal should still be to provide enough details to reproduce independent of any accompanying code base. Even with these missing details I'm a bit concerned with the results as well, and how many conclusions we can draw from what is presented.

   Q3-1. How many runs were performed? (all)
   Q3-2. What is the statistical significance of the provided results? (regression and off-policy prediction)
   Q3-3. What were the hyperparameters chosen, how were the parameters chosen? (all)
   Q3-4. What loss functions were used for the classification and regression experiments?
   Q3-5. Why not choose another type of regression problem, rather than another classification problem?
   Q3-6. What optimizer was used for training?

   On top of these missing details, some more general concerns:
   C3-1. I'm assuming we are only using a constant learning rate for all the experiments. Is this reasonable. Could we expect the results to change if the learning rate is able to take the variance of updates into account (i.e. using RMSProp). These experiments could be worthwhile pursuing, because the analysis only covers static learning rates. This would also help elucidate more about the interaction between resampling and reweighting in systems that are closer to what is being used in practice.
   C3-2. For the off-policy prediction setting. What is the discount parameter for the problem you are considering? How are you calculating the error? Only through online data? Or through sampling from the sampling a set of states from the state distribution of the behavior policy? Does this environment relate back to something in the real world that is of interest to make predictions on? What is the motivation for this problem? What are the x and y axis in the left plot of figure 5? Did you mean to calculate error using $V_\pi$? The current notation is often used for the optimal value function, which is a different concept.


Some minor corrections:
 - Page 7 "...objective is to find the value function of police \pi..."
 - Abstract "reweighting outperforms resampling when ...", you draw the opposite conclusions in the paper.

--- Updated score ---
The authors did a nice job of addressing many of my concerns. But there are still some lingering issues with the experimental design (especially with the reinforcement learning experiments). The main concern I have is why the variance for your results is so low from run to run. This could suggest the problem is too easy, or that there is a bug somewhere. While I don't think it is enough to outright reject the paper, it still puts me on the fence about a strong accept.

---

> ### Author Response · Authors · 2020-11-18
> **Response to AnonReviewer1**
>
> We would like to thank the reviewer for very careful evaluations and constructive suggestions. First of all, we hope that the reviewer can clarify the meaning of "*weight shifting*" in question 1 and "*statistical significance*" in Q3-2. We attempt to answer it based our interpretation, but it will help improve our response if the reviewer could make it clearer.
>
> **1**.
> - The off-policy evaluation problem can be reformulated as the general form we proposed in Section 2. Let the TD error at $(s,a_i)$ be $\delta_i^s(\theta) = R(s) + \gamma V_\theta(s_{t+1}|s_t = s,a_t =a_i) -V_\theta(s_t)$, then the TD error at state $s$ under policy $\pi$ should be $\delta^s_\pi(\theta) = \sum_{i=1}^{m} \delta^s_i(\theta) \pi(a_i|s)$.
>
> If one uses GTD method to do the off-policy evaluation, then the loss function is $V(\theta) = \sum_{s}(\delta^s_\pi(\theta))^2 = \sum_s\sum_{i,j=1}^{m} \delta^s_i(\theta)\delta^s_j(\theta) \pi(a_i|s)\pi(a_j|s)$. Here $\pi(a_i|s)\pi(a_j|s)$ can be viewed as the population proportion $\alpha_{ij}$ for group $(i,j)$.
>
> If the behavior policy is different from the target policy, then the empirical TD error at state $s$ is $\delta^s_\mu(\theta) = \sum_{i=1}^{m} \delta^s_i(\theta) \mu(a_i|s)$. Therefore, the empirical loss function is $\hat{V}(\theta) = \sum_{s}(\delta^s_\mu(\theta))^2 = \sum_s\sum_{i,j=1}^{m} \delta^s_i(\theta)\delta^s_j(\theta) \mu(a_i|s)\mu(a_j|s)$, where $\mu(a_i|s)\mu(a_j|s)$ can be viewed as the empirical proportion $f_{ij}$ for group $(i,j)$. This fits the form of the problem setup in Section 2.
>
> If one uses TD method to do the off-policy evaluation as what we did in the numerical experiments in Section 5, although TD method do not have an explicit loss function, the update rule is similar to SGD: $\theta_{t+1} = \theta_t + \xi(\theta_t)$, where $\xi$ is an unbiased estimate for the TD error $\delta^s_\pi(\theta_t)$. Then the population gradient is $\delta^s_\pi(\theta) = \sum_{i=1}^{m} \delta^s_i(\theta) \pi(a_i|s)$, while the empirical gradient is $\delta^s_\mu(\theta) = \sum_{i=1}^{m} \delta^s_i(\theta) \mu(a_i|s)$. The actual proportion for group $i$ is $\pi(a_i|s)$, which is equivalent to $\alpha_i$ in the setting of Section 2; while the empirical proportion of samplings for group $i$ is $\mu(a_i|s)$, which is equivalent to $f_i$ in the setting of Section 2.
>
> - The loss function of our regression is mean squared error that can be understood as the following
> $$V(\theta) = \frac{1}{N}\sum_{i}^N (y_i - g(x_i;\theta))^2 = \frac{1}{N}\sum_{s_1} (y_{s_1} - g(x_{s_1};\theta))^2 + \frac{1}{N}\sum_{s_2} (y_{s_2} - g(x_{s_2};\theta))^2 $$
> where $|\\{s_1\\}|=N_1$ data samples $(x_i, y_i)$ have $y_i<200k$ and $|\\{s_2\\}|=N_2 =N-N_1$ data samples $(x_i, y_i)$ have $y_i>400k$. The ratio $N_1/N_2 = 11767/1726$.
>
> **Q1-1**: The weights will shift in a bounded domain. The weights won't go to infinity because we assume the loss function goes to positive infinity as the weights goes to infinity.
>
> **Q1-2**: In general, deep Neural Network is an interesting setting but still too complicated to study at the first step. We tend to illustrate our theoretical findings in the cleanest fashion at this stage. NN in lazy training is approximately training in a quadratic function, which is not very interesting. We hope we could go beyond that in the future work.
>
> **Q1-3**: Lemmas 3 and 4 can be extended to loss functions with a finite number of minimizers in 1D (See Lemmas 9 and 10 in Appendix B.3 for details).  While Lemmas 5 and 6 cannot be easily extended to multiple minimizers. In fact, transition time in multi-dimension with more than two local minima is still an open question in mathematics. The Eyring-Kramers formula cannot be applied to arbitrary local minima or even any two adjacent minima. Specifically, if the loss $V$ has $n$ local minimizers $\theta_i^*, i= 1,2,\cdots,n$, there exists an ordering, $\theta_1^* \prec \cdots \prec\theta_n^*$ from deepest to shallowest, such that Kramers’ law holds only for the transition time from each $\theta_k^*$ to the set $M_{k-1} = \\{\theta_i^*\\}_{i=1}^{k-1}$. (See Section 3 in [Nils Berglund.  Kramers’ law:  Validity, derivations and generalisations, 2011.]  for more details.)
>
> **Q1-4**: We summarize the generalization of our results as follows:
> - Lemmas 1 and 2 can be extended to piecewise strictly convex functions because around all local minima, the loss function can be approximated by quadratic functions using Taylor expansions.
> - Lemmas 3 and 4 can be extended to piecewisce strictly convex functions in 1D, as well as loss functions with a finite number of local minima in 1D. Please check Remark 2 in blue after Lemma 4 and Appendix B.3 for details.
> - Lemma 6 cannot be easily generalized to multi-dimensional loss functions, since it is still an open problem in mathematics.
>
> We continue our response to Q2, Q3 in the next comment box.

---

> > ### Author Response · Authors · 2020-11-18
> > **Continuation of our response**
> >
> > **2**. We have included extensive numerical results with various learning rates in the revised paper, Appendix D.
> >
> >
> > **3**.
> > **Q3-1**: Classification: we apply a 10-fold cross validation on the training dataset. For each subset, we train the model with epochs=20 and the batch-size=1000. Nonlinear Regression: for each method, we run the experiment for 10 times. In each time, we split the data randomly and use 70% as the training data and 30% as the test data. We train the model with epochs=400 and the batch-size=32.
> > Off-policy prediction: Since the variance of the simulations are small, the results we presented in the paper are all from one single run.
> >
> > **Q3-2**:
> > - Regression: we present the mean and standard deviations of MSE for each method in Table 1 in the revised paper.
> > - Off-policy prediction: Both resampling and reweighting methods in this case have small variance for multiple runs. So we just plot the results for one single run.
> > **Q3-3**:
> > - For classification and nonlinear regression, we have stated all hyper-parameters that we use in other answers to Q3. For the classification problem, we use the original training setting from the imbalanced-learn package. For the regression problem, we choose the parameters depending on reasonable MSE outcomes.
> > - Off-policy prediction: learning rate $\eta = 0.1$ and discount factor $\gamma = 0.9$. We use the TD update based on a trajectory with length $10^5$ generated by the behavior policy.
> > **Q3-4**:
> > - Classification: loss = binary cross-entropy.
> > - Nonlinear regression: loss = mean squared error
> > **Q3-5**: We have replaced the second numerical experiment with another regression problem, please see the details in the revised paper.
> >
> > **Q3-6**: We use the Adam optimizer with default setting for both classification and regression problems.
> >
> > **C3-1**: In fact, for the classification and regression problems, instead of using SGD with a fixed learning rate, we use the adaptive learning method Adam as the optimizer to train the neural network for learning efficiency and reasonable performance. When combined with these adaptive learning methods, we can also see that resampling consistently outperforms reweighting with various sampling ratios.
> >
> > As we discussed in theory that the biased sampling ratio $f_1/f_2$ is reflected in the variance associated to reweighting, the noise of stochastic gradient algorithms makes optimal learning rate selections much more restrictive for reweighting. Even when adaptive learning rates are used, the trajectory can still be possibly trapped in bad local minima when only reweighting is used. To our knowledge, none of the available adaptive learning methods can help to find ideal local minima. We also tried different adaptive learning methods such as RMSProp for the regression problem, the outputs are similar to the Adam case:
> > MSE for Baseline: 106499.90, MSE for Resampling: 78059.81, MSE for Reweighting(1/9): 90509.12.
> >
> > **C3-2**: More details for the off-policy prediction:
> > - The discount parameter is $0.9$.
> > - In this case, the exact value function $V^*(s)$ (now change to $V^\pi(s)$) can be directly calculated from the Bellman equation if the target policy is known. Therefore, the error is directly calculated by ${e}_t = \lVert V_t(s) - V^*(s) \rVert^2_2$.
> > - The motivation of the off-policy prediction is to verify that resampling is better than reweighting in this setting, so we choose to work on a toy model with clear transition matrix. One can refer to [Schlegel et al., 2019]  for experiments with more practical problems.
> > - The x-axis is the value of the value function, and the y-axis represents the states $s = 1,\cdots, 32$.
> >
> > - Here $V(s)^*$ represents $V^{\pi(s)}$. We now change the notation to avoid confusions.

---

> > > ### Comment · AnonReviewer1 · 2020-11-25
> > > **Thank you for the clarifications**
> > >
> > > Thank you for the extensive comments and clarifications! I've found them useful and enlightening.
> > >
> > > Some clarifications of my own:
> > > - "Weight shifting": This was imprecise on my side. What I likely should have said was "Are we assuming the weights during learning will only be within a ball around these local minima." This also is answered based off the problem you are considering here, where you mention "the loss functions go to infinity as the weights increase". This was more thinking beyond the examples provided and trying to understand how this might relate to different classes of functions.
> > > - "Statistical Significance": https://www.investopedia.com/terms/s/statistical-significance.asp In brief, I mean how confident are we the empirical results are not just of random circumstance.
> > >
> > > Some more comments.
> > > I think you did a really nice job going through and making changes to the paper based on reviewer feedback. And you have addressed many of my concerns adequately (in my judgement). I have a few lingering concerns that I would like to think about here and try and get this paper to a point where I'm comfortable accepting (I do plan on increasing my score, just how much will depend on the below concern!)
> > >
> > > My lingering point of concern is "Off-policy prediction: Both resampling and reweighting methods in this case have small variance for multiple runs. So we just plot the results for one single run.". So lets try and understand/figure out what is going on. To start: Where is the randomness coming from in this experiment? Via just the sampling method? Or random initialization of weights? Somewhere else? Given how noisy the resulting error is, I'm surprised there is little variance between runs.

---

> > > > ### Author Response · Authors · 2020-11-25
> > > > **Address to your concerns**
> > > >
> > > > Thank you very much for your response.
> > > >
> > > > Sorry that we did not make it clear for the off-policy prediction.
> > > >
> > > > This experiment considers a single target policy with 3 different behavior policies. For each behavior policy, we test resampling vs. reweighing. So there are six tests in total.
> > > >
> > > > For each test, part of the noise comes from the trajectory of the behavior policy. We generate a sequence of 10^5 steps following the behavior policy. This trajectory is random since the action at each time step is randomly chosen according to the behavior policy.
> > > >
> > > > The resampling method also introduces some minor randomness since the resampling is random. The reweighting does not introduce extra randomness as the algorithm simply follows the behavior policy trajectory (but with the correct weight used for parameter updating).
> > > >
> > > > Random initialization of the NN weights/parameters introduces extra randomness as well.
> > > >
> > > > Due to the long trajectory used, the randomness tends to average out in the value function prediction. For each of the six tests, we have performed >10 runs. We have observed that the resulting value functions exhibit only minor variations. The left plot of Figure 4 already includes 6 curves (one for each test), and therefore we decided to plot only one run for each test.
> > > >
> > > > The right plot in Figure 4 does show some oscillations. However, since this plot is in the log scale, these oscillations (around -1.5 in log scale) are relatively small in terms of the linear scale in the left plot of Figure 4.

---

### Official Review · AnonReviewer2 · 2020-10-28
**Interesting analysis, but some details are missing or wrong**

**Rating:** 6
**Confidence:** 3

**Review:**

This paper provides an analysis of why resampling can be better than reweighting in some cases. By observing the behaviour of resampling and reweighting in simple optimizations with SGD, the theoretical results show that resampling tends to be more stable. The general analysis is based on SDE approximation. Experiments on classification and off-policy evaluation show that resampling can be better in some cases.

Strength:
- Simple examples and analysis showing the instability of reweighting and the possible reasons

Weakness:
- The theory is fairly restricted
- Some details are missing
- Experiments are unclear

Detail comments:
1. The assumptions of Lemma 5 and 6 are strong. In most cases, the optimization landscape is likely to have more than two local minimas. Moreover, Lemma 6 assumes that the relative error is bounded by epsilon, which is difficult to verify in practice.

2. The effect of the learning rate is not sufficiently demonstrated. The learning rate \eta plays a role in Lemma 1 and 2. Fig.1 only shows a specific eta=0.5 without explanation. It would be interesting to see how resampling and reweighting behave with different learning rates. The same applies to the example in Sec.4.

3. There are not enough details about how Fig.1(2) and Fig.2(3) are produced. Specifically, how is the resampling conducted?

4. The regression experiment is actually binary classification.

5. The TD(0) update on page 8 is wrong. The minus sign should be plus. Additionally, using \delta_\pi and \delta_\mu can be misleading as the TD error only depends on the functional form of V instead of the policies. The setting and the results are very unclear. There are n=32 states, but the x-axis of Fig.5(left) ranges from 0 to 6. All the curves are using C=1/10, so it is not clear how they differ. What do W and S stand for in the legends?

Minor
- In the proofs of Lemma 3, Z is used as a standard Gaussian RV and partition function.
- The \theta^* in Lemma 5 should be \theta^\circ.

==== Update ====
Increased score according to the revision and discussion.

---

> ### Author Response · Authors · 2020-11-18
> **Response to AnonReviewer2**
>
> We would like to thank the reviewer for valuable comments. First of all, could the reviewer clarify the meaning of "*relative error*" in comment 1? We attempt to answer it based our interpretation, but it will help improve our response if the reviewer could make it clearer.
>
> **1**. Transition time in multi-dimension with more than two local minima is still an open question in mathematics. The Eyring-Kramers formula cannot be applied to arbitrary two mimima or even any two adjacent minima. Specifically, if the loss $V$ has $n$ local minimizers $\theta_i^*, i=1,2,\cdots,n$, there exists an ordering, $\theta_1^* \prec \cdots \prec\theta_n^*$ from the deepest to the  shallowest, such that Kramers’ law holds only for the transition time from each $\theta_k^*$ to the set $M_{k-1} = \\{\theta_i^*\\}_{i=1}^{k-1}$ (see Section 3.3 in [Nils Berglund.  Kramers’ law:  Validity, derivations and generalisations, 2011.] for more details).
>
> **2**. For both toy examples in Section 3 and 4, we can run experiments with different learning rates. When the learning rate is sufficiently small, numerical experiments show that both resampling and reweighting would perform well in the sense that the trajectories stay around the local minimum rather than jumping to the other. However, in practice we don't know the optimal learning rates because we don't know how stiff the problems could be. With that in mind, resampling is a safer choice compared to reweighting when stochastic gradient methods are applied. We have included more numerical results with different learning rates in the Appendix D for both examples. From numerical results we observe, when increasing the learning rates, reweighting more likely behaves worse than resampling.
>
> **3**. The resampling strategy we conducted in Fig.1(2) and Fig.2(3) is to randomly select the subpopulation $i$ with the probability $a_i$ with replacement in each iteration, where $a_i$ is its underlying population proportion. We include this detail in the revised manuscript.
>
> **4**. Thanks for pointing this out. We have replaced the second experiment by another nonlinear regression problem trained by the neural network. Please check details in the revised manuscript that we have uploaded.
>
> **5**. We thank the reviewer for pointing out the typos in the numerical experiments for the off-policy evaluation.
> - The reviewer is correct that the TD error does not depend on the policy. We now delete the lower index $\mu, \pi$ for the definition of TD error.
> - We now change the minus sign to plus.
> - The x-axis of Fig.5(left) represents $\\{\frac{2\pi}{32}s_i\\}_{i=1}^{32}$, therefore it ranges from $0$ to $6$. We now change the x-axis to discrete state $1$ to $32$ to avoid confusions.
> - W and S (we now change it to "RW" and "RS") stands for reweighting and resampling respectively; from top to bottom $C$ should be equal to 1/10, 1/5, 2/5 respectively. We now revise the plot and add more details in the caption of Fig.5.
>
>
> *response to minor comments*
>
> We thank the reviewer for pointing out typos and parts we missed to explain, and we have modified the manuscript accordingly.

---

> > ### Comment · AnonReviewer2 · 2020-11-25
> > **Comments**
> >
> > Thank you for the explanations and additional results. I increased my score accordingly.
> >
> > I'm still concerned about the discussion on the learning rate. The learning rates in Appendix D are similar to the ones in the main text. Instead of blindly trying out different rates, it would be more informative (as R4 pointed out) to explicitly compare the "stable ranges" (Lemma 1&2) of both methods for the toy examples. A minor comment: Fig.5(c) is different from Fig.1, and Fig.6(c) is different from Fig.2(2&3), even though the same learning rate is used.

---

> > > ### Author Response · Authors · 2020-11-25
> > > **Thank you for the comments**
> > >
> > > Thank you for the reply.
> > >
> > > Fig.5(c) is different from Fig.1 because in Fig. 5 all experiments start from $\theta_0 = 1.6$ for better comparisons (we forget to mention the starting point in the caption). In Fig.1, in order to show how instable the reweighting could be, we pick $\theta_0=1.1$ for resampling and $\theta_0=2.0$ for reweighting.
> > >
> > >  Fig.6(c) is different from Fig.2(2&3) is due to the noise from SGD.

---

### Official Review · AnonReviewer5 · 2020-11-06
**Official Blind Review #5**

**Rating:** 7
**Confidence:** 3

**Review:**

This paper provides a theoretical investigation into, and comparison between, two forms of correcting biased data: reweighting, and resampling. In particular, since previous empirical analyses have suggested that reweighting performs better in practice, the author(s) provide several theoretical explanations for this discrepancy. This is a very interesting problem to consider, and I applaud the authors on their general approach and conclusions, which are novel to the best of my knowledge. The paper is also reasonably well-written too.

Unfortunately, as it is presented here, I find the paper underwhelming for two reasons.

1. Currently, the results (aside from Lemma 6, a quick corollary of a known result) are stated to hold only for very specific toy examples. This is nice enough for illustrative purposes, but I don't see why, with a little more effort, these cannot be extended to a much larger class of objectives. In particular:
(a) Why does the stability analysis in section 3 not extend readily to combinations of strongly convex functions with disjoint supports?
(b) What difficulty does it pose to have more general bimodal functions in Lemmas 3 and 4? To me it seems that the main obstacle is the non-constant diffusion coefficient in the SDE approximation. However, in one dimension at least, the Lamperti transform (which is used in Appendix C) can be used to convert this SDE into one with a constant diffusion coefficient, and the rest of the argument should go through. Am I missing something?

2. A more minor point: some of the discussion (and the title) seems to suggest that resampling is always a better choice, which is certainly not true in general. There are settings, particularly outside of deep learning, where reweighting is superior since it typically yields lower variance. More specific language is needed throughout --- this is a phenomenon that pertains primarily to stochastic gradient methods where stability/robustness is key and weights can be small. Further on this point, the author(s) briefly mention under eqn. (4) that the key reason for the discrepancy is that the variances are quite different. I think more discussion is probably needed about these differences in variance.

I enjoyed reading the paper, but these two issues (the former in particular) prevent me from recommending acceptance, as it seems it can be greatly improved for theoretical audiences with comparatively little effort. With these improvements, or some good reasoning as to why they are infeasible, I would be happy to boost my score to a 7.

Some other minor comments:

- I am somewhat confused at the value of the numerical experiments presented. I agree that they add to the paper, but my understanding from the introduction is that it was already known that resampling methods work better in these applications than reweighting. Is this not true? If experiments of this form have not been conducted in the literature previously, this is certainly worth mentioning. Otherwise, some of these could be moved to supplementary material in favour of further discussion on the theoretical results.
- Can more than two subpopulations be considered? It seems to me that the general case involving more than two subpopulations could be reduced to successive two-subpopulation problems. Is it possible to extend some of the consequences of these results, especially Lemma 6, to this setting?
- Very minor, but I'm not especially fond of the use of f_1, f_2 for the sampling proportions. Other papers discussing Langevin approximations sometimes use f to denote the potential function, which makes things a little confusing here. Can a different letter be used instead?
- Two sentences before eqn. (4): "{s}" should just be "s".
- Figures 1 and 2: Can (a),(b),(c) be used instead of (1),(2),(3), so as not to confuse with equation numbers?
- First sentence on pg. 4: "We refer..." -> "We defer...". Also "...shows that reweighting makes problems stiffer in terms of the stability criterion": maybe "reweighting can incur a more stringent stability criterion"?
- Can the size of the axis labels (not tick labels) in Figures 1 and 2 be increased?
- Last line of pg. 5: "not-Lipschitz" -> "non-Lipschitz"
- Last line on pg. 8: "extend our analysis to unsupervised learning problems": since the previous discussion has implied these results hold very generally (point 2), it might be good to provide a little more detail on what is meant by this.

---

> ### Author Response · Authors · 2020-11-18
> **Response to AnonReviewer5**
>
> We would like to thank the reviewer for carefully evaluating our paper. Here are our responses.
>
> **1(a)**. We agree that the stability analysis we have can be extended to other types of loss functions, such as a combination of strongly convex functions with disjoint supports. The results we have can be extended to more general functions that are approximated by quadratic functions using Taylor expansions near the local minima. We pick the piecewise quadratic function as our toy example, because its computation is clean and clear to present in the paper. We have extended our results to piecewise strictly convex functions, as well as loss functions with a finite number of local minima. Please check Remark 2 in blue after Lemma 4 and Appendix B.3 in the uploaded revised paper.
>
> **1(b)**. It is true that in one dimension, we can apply the Lamperti transform to obtain a new SDE with a constant coefficient for the noise. However, such a transformation may change the properties of the corresponding potential function, and one should be cautious that the usual stationary distribution analysis may not be informative anymore. In higher dimensions, it is in general technically difficult to analyze with the stationary distribution theory without a constant diffusion coefficient. We have included new comments about this in Remark 1 after Lemma 4.
>
> **2**. Thanks a lot for your suggestions.  We plan to change our title to be "Why resampling outperforms reweighting for correcting sampling biases with stochastic descents" to address the important role that stochastic gradient methods play here. Besides, in the paper, we have emphasized that the comparison of the reweighting and resampling methods is in the setting of SGD in several places, for example in abstract and main contribution. In the revision, we are more careful about the statements that we make. In fact, both resampling and reweighting can potentially do equally well with sufficiently small learning rates. However, choosing extremely small learning rates is not applicable for training efficiency. The point we want to make is that without knowing the optimal learning rate choices a priori, resampling tends to be more friendly than reweighting in the stochastic gradient setting.
>
> We have included an elaborated discussion about the differences in variance under eqn. (4).
>
> *Response to minor comments*
>
> 1. We present the numerical experiments in order to balance different reviewers' interests. Many numerical results are scattered around in different papers, and collecting some numerical experiments and represent them make our paper look more complete. More than that, our numerical experiments tune different reweighting factors to emphasize the roles of $f_1, f_2$.
>
> 2. Our Lemmas 3 and 4 can be generalized to multi-subpopulations, please see Lemma 9 and 10 in Appendix B.3 for details.
> However, Lemma 6 cannot be easily generalized to the multi-subpopulations case. The main reason is that for a loss function with more than two local minima, the Eyring-Kramers formula cannot be applied to arbitrary two local mimima or even any two adjacent local minima. Specifically, if the loss $V$ has $n$ local minimizers $\theta_i^*, i=1,2,\cdots,n$, there exists an ordering $\theta_1^* \prec \cdots \prec\theta_n^*$, from the deepest to the shallowest, such that Kramers’ law holds only for the transition time from each $\theta_k^*$ to the set $M_{k-1} = \\{\theta_i^*\\}_{i=1}^{k-1}$ (see Section 3.3 in [Nils Berglund.  Kramers’ law:  Validity, derivations and generalisations, 2011.] for more details). Transition time within multiple minimizer, especially in multi-dimensional case, is still an open question in mathematics.
>
> 3. Thanks for your suggestions and pointing out typos. We have made changes accordingly. We plan to change $f_1, f_2$ to be $\beta_1, \beta_2$ as well as the subfigures' indexing in the future, but now we keep them in order not to confuse the rest of reviewers.

---

> > ### Comment · AnonReviewer5 · 2020-11-23
> > **Thank you for responding to my concerns**
> >
> > Thank you to the authors for responding to my concerns. The additional discussion regarding variances is appreciated and I think helps highlight the differences earlier. The new theoretical results are also very welcome, and I applaud the authors for deriving these results in such a short period of time. It has taken me a little time to parse them, but I believe the arguments are sensible.
> >
> > There are now a few generalizations of Lemmas 3 and 4, but I'm not sure why these are relegated to the Appendix now. Lemmas 9 and 10 are far more general and essentially say the same thing. Why not have these in the main text with the remark commenting on the special case in Figure 2?
> >
> > I don't think a statement comparing two specific minima is actually desirable in the general case. Indeed, the result the authors state (transition time from one minima to any other) seems exactly like what one would desire. In this case, one can just compare these transition times and construct a similar scenario where the global minimum has the smallest relative exit time.
> >
> > Regardless, I'm happy enough with the changes made to address my concerns, so I will update my score to a 6 going into reviewer discussion.
> >
> > Some other minor comments:
> > - '...if we set $g_i(\theta)$ to be the anti-derivative...' — I'm assuming this fact follows from the inverse function theorem unless I'm missing something more obvious. Either way, this is probably worth mentioning.
> > - The use of boldface E and V in the proof of Lemmas 6 and 7 is very confusing, since it seems like one is taking expectation and variance of the pdf, which is nonrandom. Can these be changed to mu and sigma, or some other choices instead?
> > - Very minor point: by convention, I think the integrals in Lemmas 6 and 7 should probably have the limits reversed with a negative sign.
> > - Since the expression for the variances is now derived in the main text, they probably aren't necessary in the proof of Lemma 6.

---

> > > ### Author Response · Authors · 2020-11-24
> > > **Response to AnonReviewer5's further comments**
> > >
> > > We thank the reviewer for constructive suggestions and kind comments. If possible, what else can we make efforts to improve so that the reviewer would boost the score to be 7 (as you mentioned earlier)?
> > >
> > > Here are the response to your comments.
> > >
> > > **Comment 1.**  *Why not have Lemmas 9 and 10 in the main text with the remark commenting on the special case in Figure 2?*
> > >
> > > **Response.**  We would like to keep Lemmas 3 and 4 in the main text as they tend to illustrate our theoretical findings in the simplest form. It is consistent with the rest of the paper. But if the reviewer insists on replacing Lemmas 3 and 4 with Lemmas 9 and 10, we will be happy to present Lemmas 9 and 10 in the main text instead.
> > >
> > >
> > > **Minor comment 1.** *'...if we set to be the anti-derivative...' — I'm assuming this fact follows from the inverse function theorem unless I'm missing something more obvious. Either way, this is probably worth mentioning.*
> > >
> > > **Response.** We did not use the inverse function theorem in the paper. $g_i(\theta) = \int 1/h'_i(s)ds$ is simply the anti-derivative of $1/h'_i(s)$.
> > >
> > >
> > > **Minor comment 2.** *The use of boldface E and V in the proof of Lemmas 6 and 7 is very confusing, since it seems like one is taking expectation and variance of the pdf, which is nonrandom. Can these be changed to mu and sigma, or some other choices instead?*
> > >
> > > **Response.** Thanks for the suggestion. We now replace $\mathbb{E}, \mathbb{V}$ by $\mu, \sigma$ to avoid confusions.
> > >
> > > **Minor comment 3.**  *By convention, I think the integrals in Lemmas 6 and 7 should probably have the limits reversed with a negative sign.*
> > >
> > > **Response.**  In fact, $\theta_1<0$, so the integral's limits are in the right order.
> > >
> > > **Minor comment 4.** *Since the expression for the variances is now derived in the main text, they probably aren't necessary in the proof of Lemma 6.*
> > >
> > > **Response.** We now delete the formula for the variance in Lemma 6 and directly refer to equation (5) in Section 2.

---

> > > > ### Comment · AnonReviewer5 · 2020-11-24
> > > > **Some clarification**
> > > >
> > > > My reason for not updating the score to a 7 is due to the main “Comment 1” that the authors have now responded to. I understand that Lemmas 3 and 4 are simpler to understand (especially over Lemmas 9 and 10), but I'm personally not fond of the way that the more general results have been relegated to supplementary material. It is certainly a matter of taste, and I agree that the example considered for Lemmas 3 and 4 is illustrative and therefore important to keep in the main body. However, as a reader, I'm much less convinced by Lemmas 3 and 4 than I am Lemmas 7 and 8. In my eyes, stating the general result first and then discussing its meaning in the more specific setting makes a stronger case and a better paper. I was uncomfortable with whole-heartedly recommending the paper in this current form, but if the authors are at least willing to consider changing this for the camera-ready version, I will boost my score to a 7.
> > > >
> > > > There still appear to be improvements that could be made, e.g.
> > > > - applying Kramers' law to compare exit times away from a particular minimum in the multiple minimizer case, as the authors suggest could be done; or
> > > > - extending the Lyapunov stability results to the piecewise strongly convex case as I suggested in my initial review.
> > > > However, I'm happy to consider these as either unnecessary to the core of the paper, or outside the scope of this paper and the subject of potential future work if this will be a significant undertaking (perhaps the authors might wish to mention this as possible future work?).
> > > >
> > > > Regarding the other minor comments:
> > > > - (1) Apologies, I had meant to refer to the claim that $g_1(\theta_1) = g_2(\theta_2)$. I must be missing something obvious here then, since I'm unsure of how to prove this fact without the inverse function theorem or some other statement about $1/h'(\theta)$. Could the authors please clarify this?
> > > > - (3) Indeed, I missed this. Thank you for clarifying.

---

> > > > > ### Author Response · Authors · 2020-11-24
> > > > > **Willing to present the general results before the specific ones in the camera ready copy**
> > > > >
> > > > > Thank you for the quick reply.  If the paper is accepted, we will reorganize the presentation of Section 4 to present the general results before the specific ones.

---

### Author Response · Authors · 2020-11-19
**Manuscript updated**

Dear Area Chair and Reviewers,

We have revised our manuscript according to the reviewers' comments and suggestions. In particular, in the revised version, we have

1. Included more extensive theoretical results, Lemma 7-10 in Appendix B.3,  as well as numerical results for different learning rates in Appendix D.

2. Replaced the second experiment in Section 5 by another nonlinear regression problem.

We highlighted the revised parts in blue color in the main text.

---

### Decision · Program_Chairs · 2021-01-07
**Final Decision**

**Decision:**

Accept (Poster)

**Comment:**


The paper theoretically investigates two bias-correction methods, reweighting and resampling. It considers a very interesting problem and presents illuminating results. The paper could benefit substantially from improving the experiments so that they clearly validate the theoretical results presented.